# TACD-GRU: Time-Aware Context-Dependent Autoregressive Model for Irregularly Sampled Time Series

## Abstract

Multivariate time series data and their models are extremely important for understanding the behavior of various natural and man-made systems. Development of accurate time series models often requires capturing intricate relationships among the variables and their dynamics. Particularly challenging to model and learn are time series with irregular and sparse observations, that may arise in domains as diverse as healthcare, sensor and communication networks. In this work, we propose and study TACD-GRU, a new Time-Aware Context-Dependent Gated Recurrent Unit framework for multivariate time series prediction (or forecasting) that accounts for irregularities in observation times of individual time series variables and their dependencies. Our framework defines a novel sequential unit that is triggered by the arrival of a new observation to update its state, and a prediction module that supports time series predictions at any future time. The current prediction module consists of and combines two novel prediction models: (i) a context-based model (TACD-GRU-CONTEXT) that relies on a set of tunable latent decay functions of time and their linear combinations to support the prediction, and (ii) an attention-based model (TACD-GRU-ATTENTION) that models dependencies among variables and their most recent values using a temporal attention mechanism. Our model shows highly competitive performance when powered by both individual and combined prediction functions outperforming existing state-of-the-art (SOTA) models on both single-step and multi-step prediction tasks across three real-world datasets.

## 1 Introduction

Building predictive (or forecasting) models of multivariate time series data is crucial for understanding the dynamics of various natural and man-made systems. Typical time series models and their predictive solutions assume that the variables are observed regularly in time (with some fixed frequency), and the dependencies among time series values are then defined with respect to these regularly observed time points. However, not all real-world systems come with regularly sampled observations. Systems where the observations are spaced irregularly in time arise naturally, for example, in healthcare where the observations made for a patient are event-driven (for example, patient's laboratory tests are performed only when ordered by a physician), or sensor networks where observations and events may be missing due to failures of the underlying equipment. The challenge here is to define and learn multivariate time series models that account for interactions and dependencies among variables and observations made at different times.

Along these lines, our objective in this paper is to develop a time series model that can predict, as accurately as possible, future values of one or more random variables defining the time series from their past irregularly sampled observations. A variety of models and methods ranging from classic statistical and modern deep learning frameworks have been developed to solve this prediction problem. Early approaches replaced the irregularly observed data with regularly spaced observations by inferring the values at the regular time points. Classic statistical auto-regressive models (AR, ARMA, ARIMA) Shumway & Stoffer (2017) or latent space models, such as, Linear Dynamic Systems (LDS) Kalman (1963) models could then be applied to support prediction at future regular times. To extend the regular predictions to arbitrary future times various smoothing and interpolation

methods were deployed. In recent years, classic statistical models have been gradually replaced with various neural architectures showing improved time series prediction performance Siami-Namini et al. (2018). Existing neural models make different assumptions on how to represent the continuous-time dynamics and predictions of multivariate time series. For example, Neural ODEs Chen et al. (2018) approximate the latent dynamics with the help of expressive differential equations while, models like mTAND Shukla & Marlin (2021) use attention mechanism to capture and represent these underlying dynamics. Based on how these irregular time series representation are learned, the existing methods can be categorized into: RNN approaches Che et al. (2018); Mei & Eisner (2017), Differential Equations approaches Rubanova et al. (2019); De Brouwer et al. (2019); Chen et al. (2018); Schirmer et al. (2022); Becker et al. (2019), attention-based approaches Shukla & Marlin (2021); Chen et al. (2023) and graph-based approaches Zhang et al. (2022); Zhang et al.; Yalavarthi et al. (2024).

While current research in time series prediction models come with different modeling assumptions and respective advantages, they also leave a room for further improvements and the development of new models. More specifically, differential equation models despite being very expressive, require significantly more training time (due to calls to the numerical solver). Moreover, attention and graph-based approaches offer greater modeling flexibility and faster training times but, they are not efficient for online deployment scenarios. Motivated by this, we propose a novel approach that aims to bridge the gap between expressiveness, computational efficiency, and practical applicability. In this work, we propose a new Time-Aware Context-Dependent Gated Recurrent Unit (TACD-GRU) that sequentially updates the state in continuous time whenever observation arrives and generates a continuous time multivariate prediction function supporting time series predictions at arbitrary future time. TACD-GRU's prediction module consists of two components: i) **TACD-GRU-CONTEXT**: a new context-based prediction model that relies on a set of latent learnable exponential decay functions to model long-term temporal dependencies among the time series variables, ii) **TACD-GRU-ATTENTION**: a new local temporal attention prediction model that defines a continuous-time prediction function over most recent observations and elapsed times since these observations were made to model short-term temporal dependencies. The two prediction models are combined using a novel **Dynamic Meta-Decision Model**. It is important to highlight that the two prediction models were designed with a slightly different objective: the context-based prediction model attempts to summarize dependencies in the entire history, and hence, may miss some local temporal dependencies that are important for the prediction task. On the other hand, since attention-based model relies only on the recent set of observations, it may miss dependencies induced by multiple observations on each individual time series. As a result, the context-based and attention-based models come with complementary strengths and intuitively, their combination may better leverage their strengths, and lead to improved overall predictive performance.

Our research work makes the following key contributions:

- We introduce two new continuous-time prediction models: TACD-GRU-CONTEXT and TACD-GRU-ATTENTION for irregularly sampled time series built upon efficient Markov state update mechanism.
- We propose TACD-GRU, a prediction model combining the above two models without introducing additional hyper-parameters, leveraging their strengths to improve the modeling of the dependencies in multivariate irregularly sampled time series.
- Comprehensive evaluation of the two new prediction models and the proposed TACD-GRU model against SOTA prediction models on multiple prediction tasks on diverse real-world datasets demonstrating that our proposed model outperforms existing prediction models for irregular time series. Moreover, our empirical evaluation shows that TACD-GRU-CONTEXT demonstrates competitive performance relative to the SOTA baselines.
- We perform additional experiments to better understand the functionality of newly proposed dynamic meta-decision model. These experiments reveal two important insights: (1) it learns the expected behavior of assigning full weight to TACD-GRU-ATTENTION for the reconstruction task, and (2) it demonstrates robustness by effectively switching to the unperturbed model if perturbations at varying noise levels are made to the predictions of one model.

## 2 RELATED WORK

In the following we review existing approaches for time series prediction for irregularly sampled time series data and contrast them to our work. Our focus will be on modern neural models. We

divide the existing approaches into four subcategories: RNN-based, ODE-based, attention-based, and graph-based approaches, based on the main mechanism they rely on when making the prediction.

**RNN approaches**. RNN methods aim to leverage efficient implementation of the hidden state update to support prediction. Standard RNN-based methods, however, work on regularly sampled time series and hence require one to use interpolation schemes to convert between regular and irregular samples. An alternative approach, Neural Hawkes Process model Mei & Eisner (2017) uses tunable exponential decay functions for event time series. GRU-D Che et al. (2018) extends this to general time series prediction, operating in continuous time and updating its hidden state with each new observation. The hidden state is adjusted using tunable exponential decay functions to account for elapsed time effects. Additionally, the missing observations are inferred using decay mechanisms that revert to some target value, typically determined by the mean of the variable. The GRU-D model is particularly similar to our TACD-GRU-CONTEXT model which also relies on a set of tunable exponential decay functions to model the effect of elapsed time on the hidden state. However, most importantly, to avoid propagating estimation errors for missing observations, TACD-GRU-CONTEXT does not interpolate missing observations.

**Differential Equation approaches.** Another widely-used approach to model multivariate time series and their dependencies in continuous time relies on differential equations models of latent space dynamics. Neural ODE Chen et al. (2018) defines temporal dynamics in continuous in terms of ordinary differential equations (ODEs) approximated by a learned dynamics model. Rubanova et al. (2019) proposes to combine this Neural ODE dynamics with VAE-like architectures (referred to as LatentODE) and RNN architectures (known as ODE-RNN) to model irregularly sampled time series. One limitation of Neural ODEs is that its solution is a function of the initial condition, however, the initial condition cannot be adapted to the observed distribution. Neural Controlled Differential Equations Kidger et al. (2020); De Brouwer et al. (2019) allows the dynamics to be continuously modulated by including the input observations. The differential equation methods typically require an external numerical ODE solver, a component that can significantly prolong model's training time. Recently, Schirmer et al. (2022); Becker et al. (2019) avoid invoking numerical solvers for continuous-time dynamics by modeling latent transitions with the help of linear stochastic differential equations, that can be solved in the closed form. In contrast to these methods, TACD-GRU does not use differential equations to model the latent state dynamics, instead it relies on (a) a set of learnable exponential decay functions for modeling temporal dependencies and time series dynamics for TACD-GRU-CONTEXT component, and (b) temporal attention mechanism for last observed values for each time series and their timing for TACD-GRU-ATTENTION component. This avoids the need for deployment of computationally costly external numerical solvers.

**Alternative approaches.** Many of the recently proposed irregular time series models rely on graph neural networks and transformer architectures. mTAND model Shukla & Marlin (2021) embeds time into fixed-sized learnable vector representation, and learn relationship between irregular observed times and fixed reference time-points to arrive at a prediction in continuous time. ContiFormer Chen et al. (2023) combines Neural ODEs with the continuous-time attention mechanism to model irregular observations. T-PatchGNN Zhang et al. leverages a patch-based mechanism on univariate time series to enhance local feature extraction and use graph neural network to learn relationships among different time series. GraFITi Yalavarthi et al. (2024) first converts irregular samples to a special bipartite graph structure and cast the prediction problem as an edge weight prediction. Other graph-based temporal Zhang et al. (2022), spatiotemporal Marisca et al. (2022) and diffusion-based Tashiro et al. (2021) models have been proposed to address irregularly spaced observations. Overall, since these models lack a Markov state representation, deploying these models in an online setting may become a computational bottleneck. Primarily, because these models require to buffer the past observations to encode a newly arrived observation and reprocess all the past observations for every inference step. In contrast, both TACD-GRU-CONTEXT and TACD-GRU-ATTENTION update its previous state when a new observation arrives thus, significantly reducing the computational overhead.

## 3 METHOD

Our goal is to define and learn a model $f(\mathbf{x}_{1:t}, \boldsymbol{\tau}_{1:t}, \Delta T)$ that given a history of $t$ observations of a $D$ dimensional multivariate time series $\mathbf{x}_{1:t}$ made at times $\boldsymbol{\tau}_{1:t}$, predicts future values of time series at time $\tau_t + \Delta T$. Since the dimension of the observation history is growing in time, we approximate it with a fixed-size state model $\mathcal{H}_t$ that summarizes what is known about the process

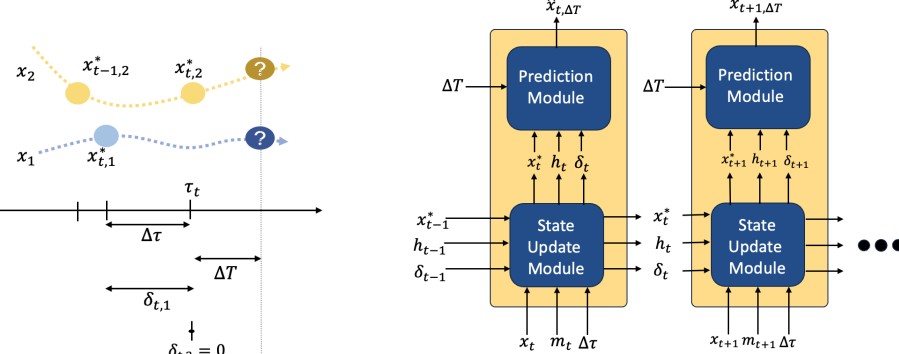

Figure 1: **(Left)** Pictorial representation of the problem setup. Consider two variables $x_1$ and $x_2$ and their dynamics (indicated by dotted lines); their discrete observations are made at irregular intervals. Arrival of an observation event prompts the update of the state $\mathcal{H}$. The prediction problem is to predict future values of variables at the time of last observation: $\tau_t$. The prediction is made at time $\tau_t + \Delta T$ for an arbitrary time horizon $\Delta T$. $x^*_{t,1}$ is the last observed value of variable $x_1$ as of time $\tau_t$; $\delta_{t,1}$ is the time-elapsed since the last observation. **(Right)** TACD-GRU cell: update and prediction. The new observation prompts the update that is followed by prediction of values $\Delta T$ into the future.

at time-step $t$. Assuming the state accurately summarizes the history of observations, it becomes Markov, making the past sequences independent of the future. A benefit of this is that the new state (after a new observation arrives) can be obtained by a state-update function that is a function of just the previous state and the new observation, simplifying greatly the application and maintenance of the models for real-time predictions. Our TACD-GRU state $\mathcal{H}_t$ consists of three components (1) a $D$-dimensional vector of last observed values up until observed time $\tau_t$: $\mathbf{x}^*_t \in \mathbb{R}^{|D|}$ comprising of last value for each variable $i$ denoted by $x^*_{t,i}$, (2) time elapsed vector since last observation $\boldsymbol{\delta}_t \in \mathbb{R}^{|D|}$; $\delta_{t,i}$ indicates the interval between $\tau_t$ and time when variable $i$ was last observed (3) a latent vector $\mathbf{h}_t$ representing dependencies among different time series values. The first two components of the state were selected to support TACD-GRU-ATTENTION prediction model, and the third component will support TACD-GRU-CONTEXT prediction model. Thus, $\mathcal{H}_t$ at time $\tau_t$ is given by:

$$\mathcal{H}_t = \{\mathbf{h}_t, \boldsymbol{\delta}_t, \mathbf{x}^*_t\}$$

Using the state model $\mathcal{H}_t$, we can estimate the future values at time $\tau_t + \Delta T$ using a new prediction function $f'(\mathcal{H}_t, \Delta T)$. We can use the new function efficiently at different time points assuming the state representation can be updated sequentially every time the new observation arrives. Let $\mathcal{H}_{t-1}$ be the representation of the state at time-step $t-1$ and $\mathbf{x}_t$ represents a new observation at time-step $t$. We want to update the state $\mathcal{H}_t$ to consistently reflect the past and the new information. Briefly, the components $\mathbf{x}^*_t$ and $\boldsymbol{\delta}_t$ of $\mathcal{H}_t$ are straightforward to update given the new observation made at $\tau_t$ and the model state calculated at the previous time-step $\mathcal{H}_{t-1}$. We use a sequential update based on RNN, more specifically GRU architecture Chung et al. (2014), to define and calculate the update of $\mathbf{h}_t$ component of the state model. Figure 1 shows how our method processes sequential inputs at each time-step.

In the following subsections, we describe the details of our prediction model, in particular, its three key components: (i) State update module, and (ii) Prediction module and (iii) Dynamic meta-decision model that combines two prediction mechanisms we include and define in TACD-GRU model. The above modules are applied at each time-step that is triggered by the arrival of a new observation.

## 3.1 State Update Module

As new observations arrive in time, the state model $\mathcal{H}_t$ representing the information seen so far, is updated with new observations. Recall that the state consists of three components: $\mathbf{x}^*_t$ - last observed values for each variable defining the multivariate time series and $\boldsymbol{\delta}_t$ - time elapsed since the last value has been observed that support TACD-GRU-ATTENTION model, and a latent vector $\mathbf{h}_t$ representing interactions among the variables and their time series that support TACD-GRU-CONTEXT model. Next we describe how each component is updated.

**TACD-GRU-ATTENTION: Update of last observed values.** The input (new observation) at each time-step $t$ is defined by an input vector $\mathbf{x}_t$ and a mask vector $\mathbf{m}_t$ indicating if an individual variable $i$ is observed at time-step $t$:

$$m_{t,i} = \begin{cases} 1, & \text{if } x_{t,i} \text{ is observed} \\ 0, & \text{otherwise} \end{cases} \qquad (1) \\ (2)$$

Given the input and mask vector, the state component representing last observed values $\mathbf{x}_t^*$ is obtained by simply replacing the old values $\mathbf{x}_{t-1}^*$ from the previous time-step with new values whenever the variable is observed at the current time-step.

$$\mathbf{x}_t^* = \mathbf{m}_t \odot \mathbf{x}_t + (\mathbf{1} - \mathbf{m}_t) \odot \mathbf{x}_{t-1}^*. \qquad (3)$$

**TACD-GRU-ATTENTION: Update of elapsed times since last observation.** The elapsed time component $\boldsymbol{\delta}_t$ of the state keeps track of time since the last observation for each variable was made. This component is updated as follows:

$$\delta_{t,i} = \begin{cases} 0, & \text{if } m_{t,i} = 1 \\ \tau_t - \tau_{t-1} + \delta_{t-1,i}, & \text{if } m_{t,i} = 0. \end{cases} \qquad (4) \\ (5)$$

Briefly, the update function simply resets the time to 0 when variable value is observed in the current step. Otherwise it adds the time difference between the previous and current timestamps ($\Delta\tau$) to the last elapsed time value.

**TACD-GRU-CONTEXT: Update of the hidden state component.** To update the hidden state component $\mathbf{h}_t$ of TACD-GRU-CONTEXT model representing long interactions among the variables and their time series with the new observation, we rely on a time-aware RNN architecture, more specifically time-aware GRU model similar to the one proposed in Chung et al. (2014). Briefly, the hidden-state update relies on a set of learnable exponential decay functions that take the previous hidden state vector, and the time difference between the previous and the new observation. The hidden state update first decays the previous hidden state on the set of trainable decay functions and then updates the state using the info in new observation. Briefly, the learnable exponential decay function for the hidden state vector is defined:

$$\boldsymbol{\gamma}(\boldsymbol{\Delta\tau}) = \exp\{-\max(\mathbf{0}, \mathbf{W}_\gamma \boldsymbol{\Delta\tau} + \mathbf{b}_\gamma)\},$$

where $\boldsymbol{\Delta\tau}$ is the time elapsed vector indicating time-interval between two consecutive time indices: $(\tau_t - \tau_{t-1}) \cdot \mathbb{1}$; where $\mathbb{1}$ is a vector containing all 1s. The learnable parameters $\mathbf{W}_\gamma, \mathbf{b}_\gamma$ govern by how much each component of hidden state needs to be decayed before it is updated by the GRU cell:

$$\mathbf{g}_t = \boldsymbol{\gamma}(\boldsymbol{\Delta\tau}) \odot \mathbf{h}_{t-1}. \qquad (6)$$

Since observations for multivariate times series may arrive at different times, the values missing in the inputs are masked. That is, at each time-step, the recurrent unit takes two vectors as the input: (1) a mask vector ($\mathbf{m}_t$) and (2) Input vector ($\mathbf{x}_t$) where missing values are substituted with zeros:

$$\mathbf{x}_t' = \mathbf{m}_t \odot \mathbf{x}_t, \qquad (7)$$

where, $\odot$ denotes the element-wise product.

Computing the hidden state ($\mathbf{h}_t$) at $\tau_t$, given $\mathbf{g}_t$, $\mathbf{m}_t$ and $\mathbf{x}_t$ involves a set of updates similar to the ones found in the standard GRU unit:

$$\mathbf{z}_t = \sigma(\mathbf{W}_z \mathbf{x}_t' + \mathbf{U}_z \mathbf{g}_t + \mathbf{V}_z \mathbf{m}_t + \mathbf{b}_z), \qquad \mathbf{r}_t = \sigma(\mathbf{W}_r \mathbf{x}_t' + \mathbf{U}_r \mathbf{g}_t + \mathbf{V}_r \mathbf{m}_t + \mathbf{b}_r),$$
$$\tilde{\mathbf{h}}_t = \tanh(\mathbf{W}_h \mathbf{x}_t' + \mathbf{U}_h(\mathbf{r}_t \odot \mathbf{g}_t) + \mathbf{V}_h \mathbf{m}_t + \mathbf{b}), \qquad \mathbf{h}_t = (1 - \mathbf{z}_t) \odot \mathbf{g}_t + \mathbf{z}_t \odot \tilde{\mathbf{h}}_t.$$

Note that $\mathbf{W}$s, $\mathbf{U}$s, $\mathbf{V}$s and $\mathbf{b}$s are learnable parameters of the model. The parameters let us fit the hidden state component and observations so that the dependencies among time series most important for the prediction are captured.

## 3.2 PREDICTION MODULE

Our main goal is to predict future time series values from past observation sequence. We use the information represented in the state $\mathcal{H}_t$ at time-step $t$ to support the predictions. We design our model

to be flexible in terms of where in the future the predictions should be made. More specifically, we use parameter $\Delta T$ to specify how far ahead in time from time-step $t$ (time $\tau_t$) we want to predict the individual time series values. Our prediction module consists of two components: TACD-GRU-CONTEXT and TACD-GRU-ATTENTION. The prediction module relies on the state $\mathcal{H}_t$ where last observed values $\mathbf{x}_t^*$ and time-elapsed since their last observation $\boldsymbol{\delta}_t$ are used by TACD-GRU-ATTENTION model, and the hidden latent state $\mathbf{h}_t$ by TACD-GRU-CONTEXT model. Both models produce an estimate of future values for all variables defining the multivariate time series at time $\tau_t + \Delta T$. Ultimately, TACD-GRU relies on meta-decision model to combine the predictions of these models to generate its final output. In the following paragraphs, we describe in more depth first the two prediction models and after that the meta-decision model.

**TACD-GRU-CONTEXT.** Our context-based prediction model, TACD-GRU-CONTEXT, relies on the decayed hidden state $\mathbf{g}_{t,\Delta T}$. Briefly, the dynamics of the hidden state component uses a set of learnable exponential decay functions that change as time progresses. The observations, when they arrive, are able to change the values of the hidden state, but in their absence, the decay functions drive the hidden state dynamics. More formally, the time series value prediction can be defined:

$$\widehat{\mathbf{x}}_{t,\Delta T}^c = F_{out}(\mathbf{g}_{t,\Delta T}). \tag{8}$$

Here, $F_{out}$ is a linear function composed of multi-layer neural network, and $\mathbf{g}_{t,\Delta T}$ is $\mathbf{h}_t$ decayed exponentially in $\Delta T$ time. The decay function is trainable, and is shared with the state update model:

$$\mathbf{g}_{t,\Delta T} = \boldsymbol{\gamma}(\boldsymbol{\Delta \tau}) \odot \mathbf{h}_t. \tag{9}$$

**TACD-GRU-ATTENTION.** Our second prediction model relies on the most recent set of observations and their timings. It aims to capture predictions and value dependencies the hidden state component may not be able to model through a fixed set of continuous time exponential decay functions and their combinations. The prediction signal here is defined by a residual signal that models a change from last observed value. Since individual time series may interact, we consider a variant of the scaled dot-product attention mechanism that combines last values of all time series and times elapsed since their last observation to predict the residual signal, that is, a change from their last value. More specifically, we predict the future time series value as:

$$\widehat{\mathbf{x}}_{t,\Delta T}^a = \mathbf{x}_t^* + \mathbf{w}_s \odot Attn(\mathbf{x}_t^*, \boldsymbol{\delta}_{t,\Delta T}) + \mathbf{b}_s. \tag{10}$$

Here, $\mathbf{w}_s, \mathbf{b}_s \in \mathbb{R}^{|D|}$ are learnable weights to scale the output of the attention module, and $\boldsymbol{\delta}_{t,\Delta T} = \boldsymbol{\delta}_t + \Delta T$ denotes elapsed times since the last observation projected to the future time where the prediction is being made. To achieve interactions amongst all last observed values and their deltas, we embed them into a fixed size vector representation, followed by scaled dot-product attention weighted summation of last observed values. Given that dot-product attention is agnostic to the pair-wise feature indices being compared, we concatenate the time-elapsed embedding with the time series embedding. This enables the dot-product operation to capture similarities across variables coupled with their respective elapsed-times. The resulting concatenated embedding are $d_k$-dimensional.

$$Attn(\mathbf{x}_t^*, \boldsymbol{\delta}_{t,\Delta T}) = Softmax(\frac{\mathbf{Q}\mathbf{K}^T}{\sqrt{d_k}})\mathbf{x}_t^*. \tag{11}$$

A query vector for $i^{th}$ time series (also represents $i^{th}$ row in $\mathbf{Q}$) is obtained by concatenating elapsed-time embedding and the time series embedding:

$$\mathbf{q}_i = \text{Concatenate}([\phi(\delta_{t,\Delta T,i}), \eta(i)]). \tag{12}$$

Since the key ($\mathbf{K}$) and query ($\mathbf{Q}$) matrices represent the same quantities and for parameter efficiency, we share the embedding function for them (i.e. $\mathbf{Q} = \mathbf{K}$). Note that the scaling parameters $\mathbf{w}_s$ and $\mathbf{b}_s$ are necessary here because, if we exclude them, assuming $\mathbf{x}_t^* \in [0,1]$, then $Attn(\mathbf{x}_t^*, \boldsymbol{\delta}_{t,\Delta T}) \in [0,1]$, and so, $\widehat{\mathbf{x}}_{t,\Delta T}^a$ is a non-decreasing function of $\mathbf{x}_t^*$. With the scaling parameters, the model is more flexible as it is also able to predict reduction from last observed value.

The elapsed-time embedding function $\phi$ converts elapsed value into a fixed $d_a$-dimensional representation using previously proposed Time2Vec model Kazemi et al. (2019). Introduction of sine non-linearity for all except the first component in $\phi$ embedding function helps us to encode periodicity in the elapsed times

$$\phi(\delta_{t,\Delta T,i})[j] = \begin{cases} \omega_0 \cdot \delta_{t,\Delta T,i} + \alpha_0, & \text{if } j = 0 \tag{13} \\ sin(\omega_j \cdot \delta_{t,\Delta T,i} + \alpha_j), & \text{if } 0 < j < d_a. \tag{14} \end{cases}$$

where, $\omega_j$ and $\alpha_j$ used to compute the $j^{th}$ component are learnable parameters. Time series indicator is embedded in $d_b$-dimensional vector as a function of its index using learnable linear embedding function $\eta$. The concatenated embedding of size $d_k (= d_a + d_b)$, is used to compute the attention weights for each variable.

**Dynamic Meta-Decision Model.** Finally, two prediction models TACD-GRU-CONTEXT and TACD-GRU-ATTENTION presented above are combined using a dynamic meta-decision model. The final output is formed by a convex combination of individual predictions, and the meta-decision model picks the weights. Briefly, a non-linear meta-decision function consisting of multi-layer perceptron (MLP) network maps the decayed hidden state to a scalar, followed by a sigmoid non-linearity to ensure $[0, 1]$ output range.

$$c_o(\mathbf{g}_{t,\Delta T}) = F_{meta}(\mathbf{g}_{t,\Delta T}) = \sigma(\text{MLP}(\mathbf{g}_{t,\Delta T})),$$

The final prediction is a convex combination of the two estimates using $c_o(\mathbf{g}_{t,\Delta T})$ as the coefficient:

$$\widehat{\mathbf{x}}_{t,\Delta T} = c_o(\mathbf{g}_{t,\Delta T}) \cdot \widehat{\mathbf{x}}_{t,\Delta T}^c + (1 - c_o(\mathbf{g}_{t,\Delta T})) \cdot \widehat{\mathbf{x}}_{t,\Delta T}^a. \tag{15}$$

We propose a dynamic weighting scheme because one estimator may outperform the other in certain scenarios. Since hidden state is the most predictive representation of the historical observations, we hypothesize that it can *encode* this information. Consequently, the dynamic meta-decision module serves as a *decoder* assigning more weight to the estimator that performs better in a given context.

### 3.3 Summary of TACD-GRU

TACD-GRU unit process that is triggered by the arrival of the observation works by first updating of state model $\mathcal{H}_t$, and then calling the prediction module. Detailed pseudocode implementation of TACD-GRU's step function and its components is presented in the Algorithm 1, Algorithm 2 and Algorithm 3. Figure 1 depicts how time-steps are processed for TACD-GRU. Briefly, new observations $(\mathbf{x}_t, \mathbf{m}_t)$ along with time-step difference $\Delta\boldsymbol{\tau}$, and the previous state model $\mathcal{H}_{t-1}$ are fed as the input at each time-step to enable the state update and prediction based on the information available up to time-step $t$.

## 4 Empirical Evaluation

We perform empirical evaluation of TACD-GRU and its components TACD-GRU-CONTEXT and TACD-GRU-ATTENTION on three irregularly sampled multivariate time series datasets on two prediction tasks: single-step and multi-step predictions (see below). We provide the description of the datasets in Appendix B and present the results in the Section 5. **Models.** We compare the proposed TACD-GRU, TACD-GRU-CONTEXT (labeled as 'TACD-GRU-C') and TACD-GRU-ATTENTION (labeled as 'TACD-GRU-A') models with the SOTA baseline models (description included in Appendix C) for irregularly sampled time series. **Model Training.** The multivariate input sequence is masked based on the specific task, such as masking the next time-step for single-step prediction. Models are trained to reconstruct the entire input sequence after observing partially masked sequence. We use Mean Squared Error (MSE) loss function to optimize the model parameters. After training, the models are evaluated on how accurately they predict the masked values. It is important to note that at any given step, the prediction horizon for the next target ($\Delta T$) is determined by the time-elapsed between the current time and the time of occurrence of the subsequent observation in the data ($\Delta\tau$). **Evaluation Criteria.** We evaluate models on MSE, Mean Absolute Error (MAE) and Win Rate on the predicted values in the test split.

## 5 Results and Discussion

We assess the performance of TACD-GRU on two forecasting tasks: single-step prediction and multi-step prediction. The evaluation is conducted across three diverse datasets derived from: United States Historical Climatology Network (USHCN), Physionet, and the Medical Information Mart for Intensive Care-III (MIMIC-III). We provide the details on these datasets in Appendix B.

**Single-step prediction.** Single-step prediction task requires the model to predict the observations made at the next time-step after having observed past sequence of observations. Table 1 summarizes the predictive performance of all the models on the single-step prediction task across the three datasets. For the USHCN dataset, ContiFormer and Latent ODE, mTAND and TACD-GRU are the best performing models in terms of MSE. TACD-GRU achieves the lowest MSE and MAE

on Physionet and MIMIC-III. In the case of USHCN dataset, TACD-GRU-ATTENTION model outperforms the TACD-GRU-CONTEXT model, suggesting that capturing short-term dependencies may be crucial for this task. In contrast, for Physionet and MIMIC-III, the relationship is reversed suggesting that more historical context is needed for accurate predictions on these datasets.

Table 1: Comparison of models on single-step prediction on USHCN, Physionet and MIMIC-III datasets. We report the mean and standard deviation of MSE ($\times 10^{-2}$) and MAE ($\times 10^{-2}$) on multiple distinct random seeds.

| Model | USHCN | | Physionet | | MIMIC-III | |
|---|---|---|---|---|---|---|
| | MSE ($\downarrow$) | MAE ($\downarrow$) | MSE ($\downarrow$) | MAE ($\downarrow$) | MSE ($\downarrow$) | MAE ($\downarrow$) |
| f-CRU | $0.020_{\pm 0.007}$ | $0.455_{\pm 0.077}$ | $1.095_{\pm 0.069}$ | $5.295_{\pm 0.132}$ | $1.191_{\pm 0.043}$ | $6.535_{\pm 0.214}$ |
| mTAND | $\mathbf{0.007}_{\pm 0.004}$ | $0.194_{\pm 0.069}$ | $0.330_{\pm 0.015}$ | $3.411_{\pm 0.130}$ | $1.260_{\pm 0.018}$ | $6.885_{\pm 0.082}$ |
| GRU-D | $0.015_{\pm 0.009}$ | $0.386_{\pm 0.110}$ | $0.672_{\pm 0.026}$ | $5.459_{\pm 0.094}$ | $0.984_{\pm 0.028}$ | $5.853_{\pm 0.102}$ |
| Latent ODE | $\mathbf{0.007}_{\pm 0.004}$ | $\mathbf{0.127}_{\pm 0.031}$ | $0.676_{\pm 0.005}$ | $5.302_{\pm 0.001}$ | $1.209_{\pm 0.018}$ | $6.490_{\pm 0.049}$ |
| ContiFormer | $\mathbf{0.005}_{\pm 0.002}$ | $\mathbf{0.125}_{\pm 0.016}$ | $0.479_{\pm 0.024}$ | $4.232_{\pm 0.001}$ | $1.348_{\pm 0.093}$ | $6.941_{\pm 0.399}$ |
| ODE-RNN | $0.019_{\pm 0.017}$ | $0.220_{\pm 0.116}$ | $0.770_{\pm 0.042}$ | $5.519_{\pm 0.273}$ | $1.429_{\pm 0.046}$ | $7.251_{\pm 0.262}$ |
| CRU | $0.030_{\pm 0.019}$ | $0.519_{\pm 0.166}$ | $0.807_{\pm 0.035}$ | $5.233_{\pm 0.242}$ | $1.236_{\pm 0.035}$ | $6.735_{\pm 0.139}$ |
| RKN-$\Delta_t$ | $0.015_{\pm 0.016}$ | $0.367_{\pm 0.178}$ | $0.680_{\pm 0.042}$ | $4.854_{\pm 0.197}$ | $1.292_{\pm 0.042}$ | $6.820_{\pm 0.081}$ |
| GRU-$\Delta_t$ | $0.035_{\pm 0.001}$ | $0.717_{\pm 0.018}$ | $0.449_{\pm 0.018}$ | $4.160_{\pm 0.154}$ | $1.414_{\pm 0.051}$ | $7.226_{\pm 0.086}$ |
| T-PatchGNN | $0.065_{\pm 0.031}$ | $0.909_{\pm 0.405}$ | $0.338_{\pm 0.032}$ | $3.259_{\pm 0.168}$ | $1.226_{\pm 0.010}$ | $6.689_{\pm 0.151}$ |
| GraFITi | $0.074_{\pm 0.015}$ | $0.673_{\pm 0.042}$ | $\mathbf{0.233}_{\pm 0.009}$ | $\mathbf{2.781}_{\pm 0.025}$ | $1.419_{\pm 0.032}$ | $7.448_{\pm 0.121}$ |
| TACD-GRU | $\mathbf{0.008}_{\pm 0.002}$ | $0.248_{\pm 0.036}$ | $\mathbf{0.232}_{\pm 0.004}$ | $\mathbf{2.773}_{\pm 0.033}$ | $\mathbf{0.578}_{\pm 0.010}$ | $\mathbf{4.419}_{\pm 0.073}$ |
| TACD-GRU-C | $0.015_{\pm 0.013}$ | $0.377_{\pm 0.187}$ | $0.261_{\pm 0.006}$ | $2.957_{\pm 0.035}$ | $0.816_{\pm 0.026}$ | $5.278_{\pm 0.036}$ |
| TACD-GRU-A | $0.009_{\pm 0.001}$ | $0.348_{\pm 0.013}$ | $0.355_{\pm 0.007}$ | $3.533_{\pm 0.058}$ | $1.019_{\pm 0.005}$ | $5.786_{\pm 0.015}$ |

Table 2: Comparison of models on multi-step prediction task on USHCN, Physionet and MIMIC-III datasets. We report the mean and standard deviation of MSE ($\times 10^{-2}$) and MAE ($\times 10^{-2}$) on multiple distinct random seeds.

| Model | USHCN | | Physionet | | MIMIC-III | |
|---|---|---|---|---|---|---|
| | MSE($\downarrow$) | MAE($\downarrow$) | MSE($\downarrow$) | MAE($\downarrow$) | MSE($\downarrow$) | MAE($\downarrow$) |
| f-CRU | $1.585_{\pm 0.022}$ | $6.987_{\pm 0.064}$ | $0.688_{\pm 0.043}$ | $5.113_{\pm 0.137}$ | $1.731_{\pm 0.050}$ | $8.434_{\pm 0.160}$ |
| mTAND | $1.593_{\pm 0.017}$ | $7.440_{\pm 0.083}$ | $0.553_{\pm 0.012}$ | $4.567_{\pm 0.057}$ | $1.779_{\pm 0.018}$ | $8.621_{\pm 0.166}$ |
| GRU-D | $1.568_{\pm 0.013}$ | $7.295_{\pm 0.171}$ | $0.785_{\pm 0.040}$ | $5.781_{\pm 0.099}$ | $1.542_{\pm 0.036}$ | $7.914_{\pm 0.136}$ |
| Latent ODE | $1.523_{\pm 0.017}$ | $7.382_{\pm 0.215}$ | $0.704_{\pm 0.013}$ | $5.503_{\pm 0.024}$ | $1.773_{\pm 0.020}$ | $8.495_{\pm 0.088}$ |
| ContiFormer | $1.569_{\pm 0.006}$ | $7.369_{\pm 0.047}$ | $0.557_{\pm 0.039}$ | $4.722_{\pm 0.276}$ | $1.471_{\pm 0.031}$ | $7.823_{\pm 0.133}$ |
| ODE-RNN | $1.724_{\pm 0.019}$ | $7.817_{\pm 0.042}$ | $0.893_{\pm 0.021}$ | $6.629_{\pm 0.109}$ | $1.645_{\pm 0.021}$ | $8.048_{\pm 0.067}$ |
| CRU | $1.403_{\pm 0.042}$ | $6.586_{\pm 0.124}$ | $0.590_{\pm 0.040}$ | $4.689_{\pm 0.169}$ | $1.768_{\pm 0.042}$ | $8.571_{\pm 0.096}$ |
| RKN-$\Delta_t$ | $1.539_{\pm 0.019}$ | $6.814_{\pm 0.063}$ | $0.875_{\pm 0.037}$ | $6.299_{\pm 0.170}$ | $1.726_{\pm 0.026}$ | $8.466_{\pm 0.089}$ |
| GRU-$\Delta_t$ | $1.701_{\pm 0.007}$ | $7.749_{\pm 0.094}$ | $0.465_{\pm 0.002}$ | $4.357_{\pm 0.051}$ | $1.831_{\pm 0.013}$ | $8.736_{\pm 0.092}$ |
| T-PatchGNN | $1.749_{\pm 0.071}$ | $8.754_{\pm 0.985}$ | $0.455_{\pm 0.007}$ | $4.253_{\pm 0.118}$ | $1.318_{\pm 0.014}$ | $7.342_{\pm 0.062}$ |
| GraFITi | $1.432_{\pm 0.010}$ | $7.199_{\pm 0.340}$ | $\mathbf{0.432}_{\pm 0.010}$ | $\mathbf{3.912}_{\pm 0.081}$ | $\mathbf{1.295}_{\pm 0.027}$ | $\mathbf{7.190}_{\pm 0.103}$ |
| TACD-GRU | $\mathbf{0.955}_{\pm 0.023}$ | $\mathbf{5.049}_{\pm 0.285}$ | $\mathbf{0.436}_{\pm 0.009}$ | $3.956_{\pm 0.056}$ | $\mathbf{1.286}_{\pm 0.028}$ | $7.041_{\pm 0.116}$ |
| TACD-GRU-C | $0.981_{\pm 0.019}$ | $5.179_{\pm 0.197}$ | $0.512_{\pm 0.011}$ | $4.288_{\pm 0.020}$ | $1.477_{\pm 0.040}$ | $7.688_{\pm 0.127}$ |
| TACD-GRU-A | $1.728_{\pm 0.061}$ | $7.960_{\pm 0.233}$ | $0.557_{\pm 0.025}$ | $4.516_{\pm 0.114}$ | $2.278_{\pm 0.001}$ | $9.173_{\pm 0.015}$ |

**Multi-step prediction.** In the multi-step prediction task, we divide the time series in time into two segments. The model observes the first segment (representing the past) to predict the observations in the second segment (representing the future). We report and compare the performance on only the predicted time-points in the second (future) segment of the sequence. For Physionet prediction task, out of the total of 48 hours, first 24 hours are observed and models are compared for prediction performance in the next 24 hours. Similarly, for the USHCN dataset, with daily samples over four years, we use the first half to predict the latter. For the MIMIC-III prediction task, models observe values of 506 variables over past 48 hours from a randomly sampled time point (so called anchor point) in the patient record. The models are compared on predicted values for observations made in the next 24 hours on 363 variables defining numerical time series defining vital signs and labs. Table 2 summarizes the predictive performance of all models on the multi-step prediction task. We note that TACD-GRU achieves the lowest MSE and MAE on USHCN and MIMIC-III while sharing the first rank with GraFITi on the Physionet. **USHCN.** We note that USHCN dataset, on average, consists of longest sequences (refer to Table 3 in the Appendix). The closest competitors to TACD-GRU on this dataset are TACD-GRU-CONTEXT, CRU and RKN-$\Delta_t$ models. This indicates that recurrent models with an explicit hidden state in general lead to better performance on this dataset. This is further supported by the fact that TACD-GRU's CONTEXT outperforms the ATTENTION model. **Physionet.** For Physionet dataset, that is inherently irregularly sampled, models GraFITi, TACD-

GRU, T-PatchGNN perform quite well indicating that both the short and the long term temporal dependencies are important to learn in this setting. Improved performance in TACD-GRU over individual models can be explained by the fact that it combines the component models with these inductive biases to formulate the final prediction. **MIMIC-III.** Among the three datasets examined, MIMIC-III most closely resembles the real-world scenario. In terms of MSE and MAE, TACD-GRU, GraFITi and T-patchGNN models are the most competitive. We further investigate two key aspects: i) whether the improvements are concentrated in a small subset of predicted variables; ii) if the enhancements are limited to specific time-spans within the predicted time horizon.

**Missing data perspective.** Prior works Singh (1997); Ramoni & Sebastiani (1997) have classified the observed missingness in the data into three categories: i) Missing Completely at Random (MCAR), and ii) Missing at Random, and iii) Not Missing at Random (NMAR). In our experiments, USHCN belongs to MCAR, where the missingness is synthetically introduced by dropping the observations uniformly at random. In contrast, the Physionet and MIMIC-III datasets are instances of NMAR, where missingness is contextual, i.e., *dependent* on observed and unobserved values. For example, in these datasets, a laboratory test might be missing until an abnormal vital sign value is observed, making the physician order the test. Our analysis suggests that while TACD-GRU remain competitive in MCAR, it outperforms all the models on NMAR. These results demonstrates that TACD-GRU is better able to model the dependencies inherent in the processes generating the missingness.

**Single-step vs. Multi-step prediction** We evaluate models on the single-step and multi-step prediction tasks, given a set of historical observations. The multi-step prediction task presents a greater challenge as it requires forecasting multiple future time points without feeding intermediate observations. The absence of intervening observations in multi-step prediction increases uncertainty and compounds model errors over time, leading to reduced accuracy relative to single-step forecasts (evident from results in Table 1 and Table 2). Notably, in NMAR single-step prediction settings, our model's improvement in MSE over the best baseline significantly exceeds that in multi-step prediction. These empirical evidences suggests that under accurate historical contexts, TACD-GRU is significantly better at capturing temporal dependencies than any other time series model considered.

**Computational cost analysis.** To understand how the irregular time series models compare in terms of training time, inference time and peak memory consumption, we provide a detailed comparison in the Appendix L. In terms of train time, as shown in Figure 10a, methods that can be parallelized over time dimension (T-PatchGNN, mTAND) are the fastest, RNN methods (TACD-GRU, GRU-D, GRU-$\Delta_t$) rank second, models with linear dynamics (RKN-$\Delta_t$, CRU, f-CRU) rank third and lastly, methods that require invoking numerical solvers (ContiFormer, Latent ODE, ODE-RNN) consume the most amount of train time. The performance trade-off is clear from Figure 10b: time-parallelizable models offer the best train times at the expense of significantly higher memory consumption. Additionally, we highlight the importance of having a Markov state representation in efficient online operation of the time series models in the Figure 10d.

**MIMIC-III Qualitative Analysis.** Examining a few MIMIC-III qualitative samples suggests that GRU-D can pick incorrect trends in scenarios when the variable to be predicted is in abnormal ranges or when variable is observed very sporadically (such as White Blood Count (WBC) lab values in MIMIC-III in the bottom plot in Figure 2). In both the cases, GRU-D will interpolate towards the empirical mean of the variable, although the actual behavior of the variable might deviate away from the mean. We plot and analyze some qualitative samples from MIMIC-III in the Appendix I and visualize learned embeddings for ABPm in the Appendix J.

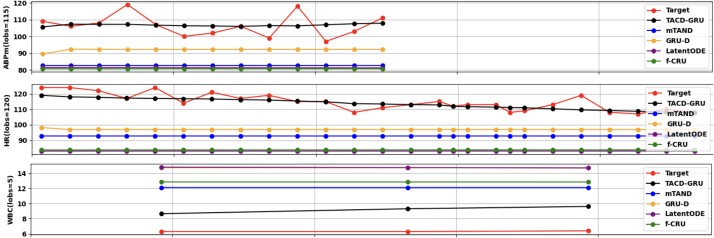

Figure 2: Qualitative samples on MIMIC-III dataset. Top two plots demonstrates that GRU-D predicts in the normal ranges (typically approximated by the empirical mean) of the variables ABPm (Mean Arterial Blood Pressure) and Heart Rate (HR), while TACD-GRU is not limited to normal range. Bottom plot is an instance of WBC prediction which is relatively more sparsely sampled.

**Examining TACD-GRU components.** Our TACD-GRU models relies on a combination of two prediction models. Two key questions that arise in this context are: ① *How good are the two models individually?* ② *Is the meta-decision model improving the prediction?* To investigate these aspects of our model, we compare the TACD-GRU with TACD-GRU-CONTEXT and TACD-GRU-ATTENTION models. We report their performance on single-step prediction task in Table 1 and multi-step prediction task in Table 2. In most cases, among the two prediction models, context based model outperforms attention based model. This is intuitive because context based model is a function of all historical observations, while attention based model only uses the last observed values. Moreover, these results suggests that the exponential decay functions are powerful enough to learn and represent the continuous temporal dynamics across multiple different datasets and prediction tasks. In both the single-step and multi-step prediction tasks, considering that TACD-GRU consistently exceeds the performance of its individual component models, it appears that these models are providing for complementary predictive value. Furthermore, this demonstrates that the meta-decision model can contextually learn to weight the two prediction models.

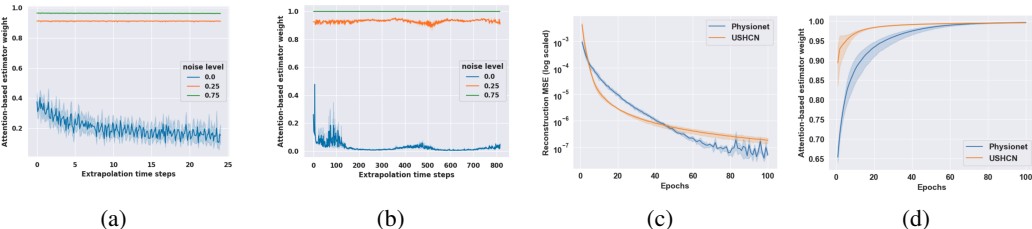

(a)  (b)  (c)  (d)

Figure 3: **Robustness.** Plots for TACD-GRU's attention based prediction model weight at different noise levels for Physionet (**3a**) and USHCN (**3b**) datasets. Both the plots indicate that as context based estimates have more noise, the meta-decision model assigns more weight to the attention model. **Reconstruction.** (**3c**) plots the reconstruction MSE over 100 training epochs for three different random seeds. (**3d**) shows the weight assigned by the meta-decision model to the attention based model in the TACD-GRU model.

**Reconstruction.** TACD-GRU is a prediction model, and recall that $\Delta T$ specifies the prediction horizon. The reconstruction task requires the model to reconstruct the current observations, which is equivalent to setting $\Delta T = 0$. Theoretically, TACD-GRU should be able to perfectly reconstruct the current observation $\mathbf{x}_t$ as it updates $\mathbf{x}_t^*$ component of the state $\mathcal{H}_t$ to include it. Empirically, for multiple datasets, we conclude that it can achieve perfect reconstruction as demonstrated in the Figure 3c. Moreover, we assess if the meta-decision model learns to focus only on TACD-GRU-ATTENTION (by assigning weight of 1) since it has sufficient information to reconstruct the current input. Interestingly, as demonstrated in the Figure 3d, the meta-decision model indeed learns this behavior quite well.

**Robustness of the meta-decision model.** In the Figure 3a and Figure 3b, we illustrate that when the TACD-GRU-CONTEXT predictions are synthetically perturbed during training for the multi-step prediction task, the decision model learns to favor the TACD-GRU-ATTENTION model by assigning it more weight. The figure demonstrates a clear trend: as the noise level increases, the meta-decision model progressively assigns more weight to the TACD-GRU-ATTENTION model. This ablation outcome, observed across multiple datasets, demonstrates the robust adaptive behavior of the meta-decision model in combining the two prediction models. We include a comprehensive description of these experiments in the Appendix K.

## 6 CONCLUSION

In conclusion, we propose and study TACD-GRU, a new recurrent neural network unit for irregularly sampled time series. Our investigation on multiple prediction tasks confirm TACD-GRU's superior performance over the existing state-of-the-art models on multiple irregularly sampled data settings. We provide extensive experiments to validate the contributions and robustness of the TACD-GRU components, further highlighting the benefits of our proposed architecture. Additionally, we introduce a new MIMIC-III derived dataset that can provide as a realistic benchmark for evaluating prediction models that handle irregular time series data.

## 7 LIMITATIONS

Unlike compared baselines such as mTAND, RKN-$\Delta_t$, f-CRU and CRU, our proposed architecture doesn't have a built-in notion of uncertainty, and incorporating it in the future model refinements of TACD-GRU remain an open challenge and our future modeling objective.

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

# APPENDIX

## A   ALGORITHMS

For more clarity, we include the exact algorithm for TACD-GRU step function in Algorithm 1 which first calls update state function followed by invoking the the prediction module.

---

**Algorithm 1** TACD-GRU Step function

---

**Input:** $\{\mathbf{x}_t, \mathbf{m}_t, \Delta\tau, \Delta T\}$, $\mathcal{H}_{t-1} = \{\mathbf{h}_{t-1}, \mathbf{x}^*_{t-1}, \boldsymbol{\delta}_{t-1}\}$
$\mathcal{H}_t = Update(\mathbf{x}_t, \mathbf{m}_t, \Delta\tau, \mathcal{H}_{t-1})$ {Call Alg. 2}
$\widehat{\mathbf{x}}_{t,\Delta T} = Predict(\mathcal{H}_t, \Delta T)$ {Call Alg. 3}
**Return:** $\mathcal{H}_t, \widehat{\mathbf{x}}_{t,\Delta T}$

---

---

**Algorithm 2** TACD-GRU Model State Update

---

**Input:** $\{\mathbf{x}_t, \mathbf{m}_t, \Delta\tau\}$, $\mathcal{H}_{t-1}$
$\mathbf{g}_t = \gamma(\boldsymbol{\Delta\tau}) \odot \mathbf{h}_{t-1}$ {decay prev. hidden state}
$\mathbf{x}'_t = \mathbf{x}_t \odot \mathbf{m}_t$ {handle missing values}
$\mathbf{x}^*_t = Update(\mathbf{x}^*_{t-1}, \mathbf{x}'_t)$ {Using Eq. 3}
$\boldsymbol{\delta}_t = Update(\boldsymbol{\delta}_{t-1}, \mathbf{m}_t, \boldsymbol{\Delta\tau})$ {Using Eq. 5}
$\mathbf{h}_t = \text{GRU}(\mathbf{x}'_t, \mathbf{m}_t, \mathbf{g}_t)$ {get updated hidden state}
**Return:** $\mathcal{H}_t$

---

---

**Algorithm 3** TACD-GRU's Prediction Module (observe $\tau_t$, predict $\tau_t + \Delta T$)

---

**Input:** $\{\mathbf{h}_t, \mathbf{x}^*_t, \boldsymbol{\delta}_t\}$, $\{\Delta T\}$
$\boldsymbol{\delta}_{t,\Delta T} = \boldsymbol{\delta}_t + \boldsymbol{\Delta T}$ {add prediction horizon}
$\mathbf{g}_{t,\Delta T} = \gamma(\boldsymbol{\Delta T}) \odot \mathbf{h}_t$ {decay hidden state}
$\widehat{\mathbf{x}}^a_{t,\Delta T} = \mathbf{x}^*_t + \mathbf{w}_s \odot Attn(\mathbf{x}^*_t, \boldsymbol{\delta}_{t,\Delta T}) + \mathbf{b}_s$ {attn. estimate}
$\widehat{\mathbf{x}}^c_{t,\Delta T} = F_{out}(\mathbf{g}_{t,\Delta T})$ {context-based estimate}
$\widehat{\mathbf{x}}_{t,\Delta T} = c_o \cdot \widehat{\mathbf{x}}^c_{t,\Delta T} + (1 - c_o) \cdot \widehat{\mathbf{x}}^a_{t,\Delta T}$ {final output}
**Return:** $\widehat{\mathbf{x}}_{t,\Delta T}$

---

## B   DATASETS

### B.0.1   USHCN

United States Historical Climatology Network (USHCN) Menne et al. (2016) is a publicly available dataset (https://data.ess-dive.lbl.gov/view/doi%3A10.3334%2FCDIAC%2FCLI.NDP019) consisting of daily measurements of 5 meteorological variables including min temperature, max temperature, precipitation, snowfall, and snow depth from 1218 observing stations across the United States. To be comparable to the results reported by Schirmer et al. (2022), we mimic their data pre-processing pipeline by: i) sub-selecting 1168 stations over a 4-year period ranging from 1990 to 1993 ii) subsample 50% of time-points to increase irregularities in time dimension; and setting unobserved rate to 20% to increase sparsity of observations, iii) 20% of the entire dataset is used for testing; we train and validate on the remaining 80% data; 25% of that is used for validation.

To be able to compare to methods listed in the Schirmer et al. (2022) for the prediction task, we replicate their: (1) pre-processing logic; (2) splitting of dataset into train, validation and test sets by using the same seed; (3) 20% partial observability in feature dimension and 50% in time dimension in TACD-GRU run; (4) masking logic for prediction task.

### B.0.2 PHYSIONET

Predicting Mortality of ICU Patients: The PhysioNet/Computing in Cardiology Challenge 2012 Silva et al. (2012) made publicly available (`https://physionet.org/files/challenge-2012/1.0.0/`) 8000 ICU patient stays that span 48 hours reporting 37 clinical real-valued time series variables observed at irregular time-intervals. The dataset includes various variables, including Non-Invasive Mean Arterial Pressure, Platelets, Sodium, and several others. We follow the pre-processing of this dataset described by Schirmer et al. (2022). Observations in time are rounded by 6 minutes. Similar to USHCN, test split consists of 20% of the data, the rest is used for training and validation; validation set consists of 25% of this split.

For fair comparison to methods listed in Schirmer et al. (2022), we ensure (1) 6-minute quantization (as done in Rubanova et al. (2019); Schirmer et al. (2022)); (2) splits are created using the same seeds; (3) same masking logic is applied for prediction task.

### B.0.3 MIMIC-III

A population cohort of 10,265 hospital admissions (on 8,799 patients) are extracted from MIMIC-III Johnson et al. (2023) (`https://physionet.org/content/mimiciii/1.4/`) based on the following criteria: i) patient record is recorded in MetaVision critical care information system, ii) the length of the patient record is between 2 and 20 days, iii) the age of the patient is between 18 and 90. From all EHR tables available in MIMIC-III, we extract (irregularly sampled) time series that include vital signs (such as Heart Rate and Mean Arterial Pressure), lab results (such as Glucose and Hemoglobin), administered medications (such as Propofol and Norepinephrine), and procedures (such as Intubation). The vital signs and lab results are numerical time series, while the rest are indicator time series, indicating if and when the event occurred.

We filter out any univariate time series that occurs less than 500 times across all patients in the cohort resulting in total of 506 time series: 393 numerical (vitals and labs), 77 medications and 36 procedures event time series. We define an *A-point*, abbreviated for Anchor-point, as a temporal moment at which a decision-making system can formulate a prediction based on the past sequence of events. We extract A-points from the filtered patient records regularly with frequency of 24 hours i.e. one sample is extracted every 24 hours from the patient record. We standardize all real-valued univariate time series (i.e. vital signs and labs) data using min-max scaling; and encode all other indicator time series as binary value 0/1. Value is 1 if event occurs; 0 otherwise. 80% of the patient hospital admission are used for training and validation (20% of train split); the rest is used for testing. Splits are constructed on disjoint patients.

Additional pre-processing is required to remove missing values encoded as 9999999 in the numerical time series. To remove outliers from univariate time series (for example, very large values of the order of $1e5$), we filter out observations that fall either in $< 0.1$ or $> 99.9$ percentile ranges. Subsampling of A-points is performed as follows:

1. For each split:

    a. For each patient admission record, filter out A-points with less than 50 events in history and prediction window.

    b. Next, randomly sample one A-point from the filtered A-points i.e. one sample per patient admission record.

2. Subsample 1000 A-points from train, 250 from validation and 200 from test set to be used for experimentation.

### B.1 DATASET ATTRIBUTES

Table 3 summarizes the mean, standard deviation, minimum and maximum of all the sequences per dataset. Importantly, the MIMIC-III dataset does not undergo any time discretization, resulting in sequence lengths spanning from 105 to 4299. In terms of average length, the order is as follows: USHCN > MIMIC-III > Physionet.

Table 3: Sequence length statistics and number of target variables across datasets

| Dataset | mean $\pm$ s.d. | min | max | # target variables |
|---|---|---|---|---|
| USHCN | $730.0 \pm 0.00$ | 730 | 730 | 5 |
| Physionet | $72.16 \pm 20.93$ | 1 | 185 | 37 |
| MIMIC-III | $244.30 \pm 288.85$ | 105 | 4299 | 393 |

## C   BASELINES

We include the following baseline methods to compare against our proposed model TACD-GRU:

**GRU-$\Delta_t$:** We consider recurrent models GRU-$\Delta_t$ as our baselines in the comparison. Gated Recurrent Unit Chung et al. (2014) have been proposed to model sequences using a set of parametric update equations. Since GRU is not time-aware, a variant of GRU that also feeds in time elapsed (since the last time-step) along with the input GRU-$\Delta_t$ is used in the comparison.

**RKN-$\Delta_t$:** Recurrent Kalman Networks (RKN) Becker et al. (2019) have been proposed to incorporate uncertainty in time series modeling. Similar to GRU-$\Delta_t$, we include the baseline RKN-$\Delta_t$ that includes time elapsed as an additional input to the model.

**GRU-D:** We compare our method to Gated Recurrent Unit-Decay (GRU-D) Che et al. (2018) that uses a mean-reverting imputation function for missing variables; applies learnable exponential decays in the input and latent dimension to account for irregular observed times.

**mTAND:** We evaluate our performance against the Encoder-Decoder generative model, Multi-Time Attention Network (mTAND-Full) Shukla & Marlin (2021), which defines reference points and represents continuous time-points with learnable embeddings to encode their relationship and generate predictions.

**CRU and f-CRU:** We consider Continuous Recurrent Units (CRU) and fast-Continuous Recurrent Units (f-CRU) Schirmer et al. (2022) as our baselines for prediction tasks. f-CRU is a fast implementation of jointly proposed CRU method. CRU consists of an encoder-decoder framework where the hidden state progression is governed by linear stochastic differential equation that allow incorporating arbitrary time-intervals between observations.

**ODE-RNN and Latent ODE:** In our analysis, we consider Latent ODE and ODE-RNN proposed in Rubanova et al. (2019) as comparative baselines. ODE-RNN model consist of latent states that adhere to ODE between observations and are updated at observations using standard RNN update equations. Latent ODE model adopts variational auto-encoder framework wherein the hidden state posterior is modeled by an ODE-RNN model. We used the configuration of Latent ODE with ODE-RNN encoder.

**ContiFormer**: We incorporate ContiFormer Chen et al. (2023) as one of our baselines for the prediction tasks. It builds upon the original transformer architecture by first extending the input irregular data to the continuous-time latent representation by assuming that the underlying dynamics are governed by the ODEs.

**T-PatchGNN**: We include T-PatchGNN Zhang et al. as one of our baselines for the prediction tasks. T-PatchGNN first segments each time series into patches of uniform temporal resolution followed by the transformer and time-adaptive GNNs to capture dependencies in the multivariate time series. We use the official T-PatchGNN code made publicly available here: `https://github.com/usail-hkust/t-PatchGNN` in our experiment pipeline.

**GraFITi**: We add GraFITi Yalavarthi et al. (2024) as one of our baselines for comparisons on the single-step and multi-step prediction tasks. GraFITi casts the time series prediction task in terms of edge weight prediction problem after converting the time series to a sparse graph structure. To incorporate this method, we make use of the official implementation available here: `https://github.com/yalavarthivk/GraFITi`.

## D    GENERIC HYPER-PARAMETERS FOR PREDICTION TASKS

The following hyper-parameter are applicable broadly across our prediction experiments:

- For all models, we use Adam optimizer Kingma & Ba (2015)
- For all models, we apply exponential learning rate decay of 0.99 and perform gradient clipping using max $l^2$-norm=1.
- For all TACD-GRU configurations, for simplicity, we experiment with the same number of time and the variable embedding sizes i.e. $d_a = d_b$ to reduce number of iterations of hyper-parameter tuning. Results can perhaps be further improved with task-specific different embedding sizes for event and time.
- For all TACD-GRU configurations, we use the same architecture for meta-decision model $F_{meta}(\mathbf{g}_{t,\Delta T})$:
  Sequential([Linear(hidden_dim, hidden_dim), ReLU(), Linear(hidden_dim,1)]).
- For all TACD-GRU configurations, we use the same architecture for the hidden to observation function $F_{out}(\mathbf{g}_{t,\Delta T})$: Linear(hidden_dim, target_dim).

## E    HYPER-PARAMETERS FOR PREDICTION TASKS ON USHCN

We keep batch size fixed to 50, the number of training epochs to 100, learning rate decay to 0.99, with gradient clipping and perform a hyper-parameter search for each model as follows:

### E.1    MTAND

For mTAND, we perform a grid search over time embedding dimension = $\{32, 64, 128\}$, latent state dimension =$\{8, 10, 16, 20\}$, number of reference points=$\{32, 64, 128\}$ and learning rate =$\{0.1, 0.05, 0.01, 0.001\}$. Of which, latent state dimension=8, number of reference points=32 and learning rate of 0.01 performs the best on the validation data.

### E.2    GRU-D

For GRU-D, we perform a search over latent state dimension = $\{8, 10, 16, 20\}$ and learning rate = $\{0.1, 0.05, 0.01, 0.001\}$. We find that configuration with latent state dimension=20 and learning rate=0.01 performs best on the validation set.

### E.3    CRU

Fixed hyper-parameters for CRU are: variance activation for encoder='square', decoder='exp', transition='relu' encoder variance activation='square', decoder variance activation='exp', number of basis matrices=20, and the same encoder and decoder network architecture as used in Schirmer et al. (2022). We perform a search on latent state dimension=$\{8, 10, 16, 20\}$ and learning rate = $\{0.1, 0.05, 0.01, 0.001\}$. We report that the latent dimension=10 and learning rate=0.05 performs the best on validation set.

### E.4    F-CRU

Fixed hyper-parameters for f-CRU include: variance activation for encoder='square', decoder='exp', transition='relu' encoder variance activation='square', decoder variance activation='exp', number of basis matrices=20, and the same encoder and decoder network architecture as used in Schirmer et al. (2022). We perform a search on latent state dimension = $\{8, 10, 16, 20\}$ and learning rate = $\{0.1, 0.05, 0.01, 0.001\}$. We report that the latent dimension=10 and learning rate=0.05 performs the best on the validation set.

### E.5    LATENT ODE

We use Latent ODE model with ODE-RNN encoder. We perform a grid search on latent state dimension=$\{8, 10, 16, 20\}$, recognition network dimension=$\{16, 32, 64\}$, number of GRU

units=$\{16, 32, 64\}$, number of generation layers=$\{2, 3\}$, number of recognition layers=$\{2, 3\}$ and learning rate= $\{0.1, 0.05, 0.01, 0.001\}$. The configuration that performs the best on validation split with latent state dimension, recognition network dimension, number of GRU units set to 20; learning rate of 0.01 and generation and recognition network layers set to 3.

### E.6 ODE-RNN

For ODE-RNN, we use GRU as the RNN model and use the adjoint solver method implemented in the library: `https://github.com/rtqichen/torchdiffeq` for solving the ODEs in a differentiable manner. For ODE-RNN, we search over latent state dimension=$\{8, 10, 16, 20\}$ and the learning rates=$\{0.1, 0.05, 0.01, 0.001\}$. We find that the configuration of latent state dimension=20 and learning rate = 0.01 works the best on the validation set.

### E.7 CONTIFORMER

We use the ContiFormer implementation released by the authors `https://github.com/microsoft/SeqML/tree/main/ContiFormer` in our implementation. Note that since USHCN has highest average sequence length, and ContiFormer is memory intensive, we can only fit a batch size of 4 samples in our GPU memory. Keeping other parameters fixed, we vary the latent state dimension = $\{8, 10, 16, 20\}$ and learning rate = $\{0.1, 0.05, 0.01, 0.001\}$. Our experiments show that latent state dimension=16 and learning rate=0.001 achieves the best performance on validation set.

### E.8 GRU-$\Delta_t$

For GRU-$\Delta_t$, we search over latent state dimensions of the GRU =$\{8, 10, 16, 20\}$ and learning rates=$\{0.1, 0.05, 0.01, 0.005, 0.001\}$. We find latent state dimension=16 and learning rate=0.005 to be the best performing configuration.

### E.9 RKN-$\Delta_t$

We use the RKN-$\Delta_t$ implementation made available by the authors `https://github.com/ALRhub/rkn_share.git`. Our RKN-$\Delta_t$ implementation uses the same encoders and decoders architecture as the CRU model. Keeping other parameters fixed, we search over latent state dimensions = $\{8, 10, 16, 20\}$ and learning rates = $\{0.1, 0.05, 0.01, 0.005, 0.001\}$. Latent state dimension=20 and learning rate=0.001 results in the best performing model.

### E.10 T-PATCHGNN

We perform a grid search over learning rates = $\{0.1, 0.05, 0.01, 0.005, 0.001\}$, time and node embedding dimensions = $\{4, 8, 16\}$, number of patches=$\{2, 4\}$ (more number of patches results in GPU OOM issue), and latent state dimension = $\{4, 8, 10, 12, 16\}$, while fixing the number of heads in one transformer layer = number of transformer layers = 1. We find that the configuration with learning rate=0.001, time and node embedding dimension=8, number of patches=2, latent state dimension=16 results in the best validation MSE.

### E.11 GRAFITI

We perform a grid search over learning rates = $\{0.1, 0.05, 0.01, 0.005, 0.001\}$, latent state dimension = $\{4, 8, 16, 20\}$, number of layers = $\{1, 2, 4\}$ and number of attention heads = $\{1, 2, 4\}$. We report that the configuration with lr=0.01, latent state dimension=8, number of layer=2 and number of attention heads=1 results in the best validation MSE.

### E.12 TACD-GRU

We perform a grid search over latent state dimensions=$\{8, 10, 16, 20\}$ (to be comparable to other baselines considered), and embedding sizes $\{2, 4, 8\}$ (smaller embedding sizes as USHCN has only 5 variables) with learning rates $\{0.1, 0.05, 0.01, 0.005, 0.001\}$. The best model configuration that

maximizes validation prediction MSE uses embedding dimensions=2, learning rate=0.1 and hidden state dimension=20.

### E.13 TACD-GRU-ATTENTION

For the prediction tasks, we report the results for attention-only ($\widehat{\mathbf{x}}^a$-only) model by using the same TACD-GRU configuration but, without the context based prediction model and meta-decision model components. So for USHCN, we use learning rate=0.1 and embedding size=2.

### E.14 TACD-GRU-CONTEXT

For the prediction tasks, we report the results for context-only ($\widehat{\mathbf{x}}^c$-only) model by using the same TACD-GRU configuration but, without the attention based prediction model and meta-decision model components. For USHCN, we use learning rate=0.1 and latent state dimension=20.

## F  HYPER-PARAMETERS FOR PREDICTION TASKS ON PHYSIONET

We keep the following hyper-parameters constant across all methods: batch size=100, number of training epochs=100, learning rate decay=0.99 and gradient clipping enabled. Below are the model specific experiments we carried out.

### F.1  MTAND

For mTAND, we perform a grid search over time embedding dimension = $\{32, 64, 128\}$, latent state dimension =$\{8, 10, 16, 20, 22, 24\}$, number of reference points=$\{32, 64, 128\}$ and learning rate = $\{0.1, 0.05, 0.01, 0.001\}$. Of which, time embedding dim=32, latent state dimension=22, number of reference points=64 and learning rate=0.01 performs the best on the validation data.

### F.2  GRU-D

For GRU-D, we perform a search over latent state dimension = $\{8, 10, 16, 20, 22, 24\}$ and learning rate = $\{0.1, 0.05, 0.01, 0.001\}$. We find that configuration with latent state dimension=16 and learning rate=0.01 performs best on the validation set.

### F.3  F-CRU

Fixed hyper-parameters for f-CRU include: variance activation for encoder='square', decoder='exp', transition='relu' encoder variance activation='square', decoder variance activation='exp', number of basis matrices=20, and the same encoder and decoder network architecture as used in Schirmer et al. (2022). We perform a search on latent state dimension = $\{8, 10, 16, 20\}$ and learning rate = $\{0.1, 0.05, 0.01, 0.001\}$. We report that the latent dimension=16 and learning rate=0.001 performs the best on the validation set.

### F.4  CRU

Fixed hyper-parameters for CRU are: variance activation for encoder='square', decoder='exp', transition='relu' encoder variance activation='square', decoder variance activation='exp', number of basis matrices=20, and the same encoder and decoder network architecture as used in Schirmer et al. (2022). We perform a search on latent state dimension=$\{8, 10, 16, 20, 32\}$ and learning rate = $\{0.1, 0.05, 0.01, 0.005, 0.001\}$. We report that the latent dimension=32 and learning rate=0.005 performs the best on validation set.

### F.5  LATENT ODE

We use Latent ODE model with ODE-RNN encoder. We perform a grid search on latent state dimension=$\{8, 10, 16, 20, 32\}$, recognition network dimension=$\{16, 32, 64\}$, number of GRU units=$\{16, 32, 64\}$, number of generation layers=$\{2, 3\}$, number of recognition layers=$\{2, 3\}$ and

learning rate= $\{0.1, 0.05, 0.01, 0.005, 0.001\}$. The configuration that performs the best on validation split with latent state dimension, recognition network dimension, number of GRU units set to 32; learning rate=0.005 and generation and recognition network layers set to 3.

### F.6 ODE-RNN

For ODE-RNN, we search over latent state dimension=$\{8, 10, 16, 20\}$ and the learning rates=$\{0.1, 0.05, 0.01, 0.005, 0.001\}$. We find that the configuration of latent state dimension=16 and learning rate = 0.001 works the best on the validation set.

### F.7 CONTIFORMER

For ContiFormer, we perform a search over the latent state dimension = $\{8, 16, 20, 32\}$ and learning rate = $\{0.1, 0.05, 0.01, 0.001\}$. Our experiments show that latent state dimension=16 and learning rate=0.001 achieves the best performance on the validation set.

### F.8 GRU-$\Delta_t$

For GRU-$\Delta_t$, we search over latent state dimensions of the GRU =$\{8, 10, 16, 20, 24, 32\}$ and learning rates=$\{0.1, 0.05, 0.01, 0.005, 0.001\}$. We find latent state dimension=32 and learning rate=0.005 to be the best performing configuration.

### F.9 RKN-$\Delta_t$

Our RKN-$\Delta_t$ implementation uses the same encoders and decoders architecture as the CRU model. Keeping other parameters fixed, we search over latent state dimensions = $\{8, 10, 16, 20, 24, 32\}$ and learning rates = $\{0.1, 0.05, 0.01, 0.005, 0.001\}$. Latent state dimension=32 and learning rate=0.001 results in the best performing model.

### F.10 T-PATCHGNN

We perform a grid search over learning rates = $\{0.1, 0.05, 0.01, 0.005, 0.001\}$, time and node embedding dimensions = $\{4, 8, 16\}$, number of patches=$\{5, 10, 20\}$, and latent state dimension = $\{4, 8, 16\}$, while fixing the number of heads in one transformer layer = number of transformer layers = 1. We find that the configuration with learning rate=0.001, time and node embedding dimension=8, number of patches=10, latent state dimension=8 results in the best validation MSE.

### F.11 GRAFITI

We perform a grid search over learning rates = $\{0.1, 0.05, 0.01, 0.005, 0.001\}$, latent state dimension = $\{4, 8, 10, 16, 20\}$, number of layers = $\{1, 2, 4\}$ and number of attention heads = $\{1, 2, 4\}$. We report that the configuration with lr=0.005, latent state dimension=16, number of layer=2 and number of attention heads=1 results in the best validation MSE.

### F.12 TACD-GRU

We perform a search over learning rate=$\{0.1, 0.025, 0.05, 0.01, 0.005\}$, the hidden state dimensions= $\{10, 16, 20, 24, 32\}$ (to be comparable to other models) and embedding dimensions = $\{32, 64, 128\}$. We find that hidden state dimension=20, learning rate=0.025, and embedding size=64 results in best validation MSE.

### F.13 TACD-GRU-ATTENTION

For the prediction tasks, we report the results for attention-only ($\widehat{\mathbf{x}}^a$-only) model by using the same TACD-GRU configuration but, without the context based prediction model and meta-decision model components. So for Physionet, we use learning rate=0.025 and embedding size=64.

### F.14  TACD-GRU-CONTEXT

For the prediction tasks, we report the results for context-only ($\widehat{\mathbf{x}}^c$-only) model by using the same TACD-GRU configuration but, without the attention based prediction model and meta-decision model components. For Physionet, we use learning rate=0.025 and latent state dimension=20.

## G  HYPER-PARAMETERS FOR PREDICTION TASKS ON MIMIC-III

We keep the following hyper-parameters constant across all methods: batch size=1 (to be able to handle the longer sequences in the MIMIC-III dataset within GPU memory constraints), number of training epochs=20, learning rate decay=0.99 and gradient clipping enabled. Below are the model specific experiments we carried out.

### G.1  MTAND

Based on MIMIC-III experiments in Shukla & Marlin (2021), we keep the following hyper-parameters fixed: time embedding dimension=128. We perform grid search on the hidden state dimension = $\{16, 32, 64\}$, encoder hidden dimension = $\{16, 32, 64\}$, the number of reference points = $\{64, 95\}$ and the learning rates=$\{0.01, 0.005, 0.001\}$. Note that for both hidden state dimensions as 64 and number of reference points as 95, we hit the memory limit on our GPU for batch size=1. Nonetheless, the resulting number of parameters (=174K) for this configuration is higher than that of our proposed model. Our validation results show that latent state dimension=64, number of reference points=95 and learning rate=0.001 performs the best across all combinations.

### G.2  GRU-D

We perform grid search over hidden state dimension=$\{16, 32, 64\}$ and learning rates=$\{0.01, 0.005, 0.001\}$. We report that hidden state=32 and learning rate=0.001 performs the best on validation set and use it to report the final results.

### G.3  LATENT ODE

We use Latent ODE model with ODE-RNN encoder. We perform a grid search on latent state dimension=$\{16, 32, 64\}$, recognition network dimension=$\{16, 32, 64\}$, number of GRU units=$\{16, 32, 64\}$, number of generation layers=$\{2, 3\}$, number of recognition layers=$\{2, 3\}$ and learning rate= $\{0.01, 0.005, 0.001\}$. The configuration that performs the best on validation split with latent state dimension, recognition network dimension, number of GRU units set to 32; learning rate of 0.001 and generation and recognition network layers set to 3. Other hyper-parameters that were kept fixed are: batch size=1, learning rate decay=0.99 with gradient clipping.

### G.4  F-CRU

We perform a grid search over the latent state dimensions= $\{16, 32, 64\}$ and learning rates=$\{0.01, 0.005, 0.001\}$. We set latent observation dimension as half the size of latent state dimension. Number of basis matrices = 20, and Gradient clipping enabled. Encoder consists of 3 ×( FullyConnected(50) + ReLU + Layer normalization) followed by linear output for latent observation and output; square activation for latent observation variance. Decoder consists of 3 × (FullyConnected(50) + ReLU + Layer normalization) followed by a linear output. Decoder output variance consists of (FullyConnected(50) + ReLU + Layer normalization) followed by linear output and square activation. Activation function for transition function is ReLU. After performing the grid search, the best configuration of hyper-parameters are: latent state dimension=64, and learning rate=0.001.

### G.5  CRU

Fixed hyper-parameters for CRU are: variance activation for encoder='square', decoder='square', transition='relu', number of basis matrices=20, and the same encoder and decoder network architecture used in f-CRU (above). We perform a search on latent state dimension=$\{16, 32, 64\}$ and learning

rate = $\{0.1, 0.01, 0.001, 0.0001\}$. We report that the latent dimension=32 and learning rate=0.0001 performs the best on validation set.

### G.6 ODE-RNN

For ODE-RNN, we search over latent state dimension=$\{16, 32, 64\}$ and the learning rates=$\{0.1, 0.05, 0.01, 0.005, 0.001\}$. We find that the configuration of latent state dimension=32 and learning rate = 0.005 works the best on the validation set.

### G.7 CONTIFORMER

For ContiFormer, we perform a search over the latent state dimension = $\{16, 32, 64\}$ and learning rate = $\{0.1, 0.05, 0.01, 0.001\}$. Our experiments show that latent state dimension=64 and learning rate=0.001 achieves the best performance on the validation set.

### G.8 GRU-$\Delta_t$

For GRU-$\Delta_t$, we search over latent state dimensions of the GRU =$\{16, 32, 64\}$ and learning rates=$\{0.1, 0.05, 0.01, 0.005, 0.001\}$. We find latent state dimension=16 and learning rate=0.001 to be the best performing configuration.

### G.9 RKN-$\Delta_t$

Our RKN-$\Delta_t$ implementation uses the same encoders and decoders architecture as the CRU model. Keeping other parameters fixed, we search over latent state dimensions = $\{16, 32, 64\}$ and learning rates = $\{0.001, 0.0005, 0.0001, 0.00005\}$. Note that we search over smaller values of the learning rate because, the model would not converge for higher ones (we get "NaN" during optimization for higher rate). Latent state dimension=32 and learning rate=0.00005 results in the best performing model.

### G.10 T-PATCHGNN

We perform a grid search over learning rates = $\{0.1, 0.05, 0.01, 0.005, 0.001\}$, time and node embedding dimensions = $\{4, 8, 16\}$, number of patches=$\{2, 4\}$ (more number of patches results in GPU OOM issue), and latent state dimension = $\{4, 8, 10, 12\}$, while fixing the number of heads in one transformer layer = number of transformer layers = 1. We find that the configuration with learning rate=0.001, time and node embedding dimension=20, number of patches=2, latent state dimension=12 results in the best validation MSE.

### G.11 GRAFITI

We perform a grid search over learning rates = $\{0.1, 0.05, 0.01, 0.005, 0.001\}$, latent state dimension = $\{16, 32, 64\}$, number of layers = $\{1\}$ (more number of layers on MIMIC-III causes OOM) and number of attention heads = $\{1\}$. We report that the configuration with lr=0.005, latent state dimension=64, number of layer=1 and number of attention heads=1 results in the best validation MSE.

### G.12 TACD-GRU

After searching over hidden state=$\{10, 16, 20\}$, learning rates=$\{0.1, 0.05, 0.01, 0.005, 0.001\}$ and embedding dimension=$\{32, 64, 128\}$, we find the combination of 16 and 64 perform the best. We set number of hidden units=16, learning rate=0.001, embedding size=64 and training for 20 epochs.

### G.13 TACD-GRU-ATTENTION

For the prediction tasks, we report the results for attention-only ($\hat{\mathbf{x}}^a$-only) model by using the same TACD-GRU configuration but, without the context based prediction model and meta-decision model components. So for MIMIC-III, we use learning rate=0.001 and embedding size=64.

### G.14 TACD-GRU-CONTEXT

For the prediction tasks, we report the results for context-only ($\widehat{\mathbf{x}}^c$-only) model by using the same TACD-GRU configuration but, without the attention based prediction model and meta-decision model components. For MIMIC-III, we use learning rate=0.001 and latent state dimension=16.

## H  EFFECTS OF TIME NORMALIZATION

Table 4: Comparison of TACD-GRU and its variants when time is normalized to $[0, 1]$ range.

| Model | Multi-step prediction MSE ($\times 10^{-2}$) | | |
| --- | --- | --- | --- |
| | USHCN | Physionet | MIMIC-III |
| TACD-GRU | $0.955 \pm 0.023$ | $0.440 \pm 0.011$ | $1.291 \pm 0.037$ |
| TACD-GRU-C | $0.981 \pm 0.019$ | $0.512 \pm 0.011$ | $1.463 \pm 0.046$ |
| TACD-GRU-A | $1.918 \pm 0.207$ | $0.557 \pm 0.025$ | $2.278 \pm 0.001$ |
| TACD-GRU-NT | $\mathbf{0.944} \pm 0.019$ | $0.455 \pm 0.014$ | $\mathbf{1.175} \pm 0.015$ |
| TACD-GRU-C-NT | $0.968 \pm 0.030$ | $0.515 \pm 0.017$ | $1.456 \pm 0.016$ |
| TACD-GRU-A-NT | $1.872 \pm 0.210$ | $0.581 \pm 0.038$ | $\mathbf{1.230} \pm 0.004$ |

As an extension to the proposed TACD-GRU model, we compare the performance of TACD-GRU and its components for the normalized and un-normalized time values. Briefly, the time is normalized using min-max scaling so that the resulting range of time is in $[0, 1]$. In terms of implementation, this requires one to compute the min and max time values in a retrospective dataset and normalizing time values before calling the TACD-GRU's forward method. In Table 4, we report and compare the TACD-GRU's performance for un-normalized and normalized time values (normalized time version with suffix "-NT"). We ensure that the hyper-parameter configuration of normalized and un-normalized times are the same including the random seeds for each experiment.

For MIMIC-III, the time granularity is in seconds, and so the time values are of the order of $10^6$ (since 1 day= $24 \times 60 \times 60$ seconds); for Physionet the time unit is in hours and so of the order of $10^1$; USHCN is daily over several years so of the order of $10^3$. It is interesting to note that the attention-based predictions see significant improvements with time normalization on MIMIC-III; small benefits on USHCN and a minor decline on Physionet. The context-based predictions show small-to-no improvements across all datasets. Overall conclusion is that the normalized-time TACD-GRU outperforms the un-normalized on datasets with higher magnitude of time values.

By restricting the time values to be in $[0, 1]$ range, we enforce that the time-elapsed value be bounded by 1. Even though the embedding function can re-scale the time-elapsed because of linear mapping in Equation (13) (using a multiplicative weight and an additive bias term), these results suggests that re-scaling time-elapsed values before embedding them, results in learning a better predictive model, espeically as the magnitude of time-elapses increase. Our hypothesis is that the learning of improved time embedding representations is due to the decreased variability in time-elapses with the help of min-max re-scaling.

## I  MIMIC-III QUALITATIVE EXAMPLES

We include qualitative examples for the MIMIC-III multi-step prediction task in the Figure 4,Figure 5,Figure 6 and Figure 7. Recall that for MIMIC-III, the prediction horizon is the future 24-hours. Given that it is difficult to display all 393 predicted numerical variables in a compact manner, we have selectively chosen to display commonly recognized univariate time series from vital signs and lab values. From top to bottom, we consider vital signs: Heart Rate (HR), Respiration Rate (RR), Arterial Blood Pressure Mean (ABPm), Non-Invasive Blood Pressure Mean (NBPm). Followed by lab values: White Blood Cell (WBC), Creatinine, Platelets, Sodium (Na), Haemoglobin (Hgb). For reference, we also provide the last observed value (labeled as "lobs" on the y-axis label) of the time series that is observed by the model. If the last value for that time series is not observed, we use "nan" to represent it. All models that we report in Table 2 are plotted along with the ground truth time series labeled as "Target".

The plots reveal that the models learn to predict approximately constant predictions in the 24 hour window for mTAND, f-CRU and Latent ODE models. We hypothesize that this could be due to limited number of training samples, limited number of training epochs. TACD-GRU and GRU-D models do produce trends in the predictions. From the figures, it appears that the TACD-GRU fit tends to result in a lower MSE compared to the rest. In general, we note that GRU-D and TACD-GRU have similar prediction profiles. However, as seen in Figure 4, there is a noticeable difference between GRU-D and TACD-GRU for ABPm. This discrepancy can be attributed to the fact that GRU-D tends to make predictions within the normal operating range (attributed to the population mean interpolation model), which in the case of ABPm is 70-100 mmHg, while TACD-GRU predicts values in the abnormal range.

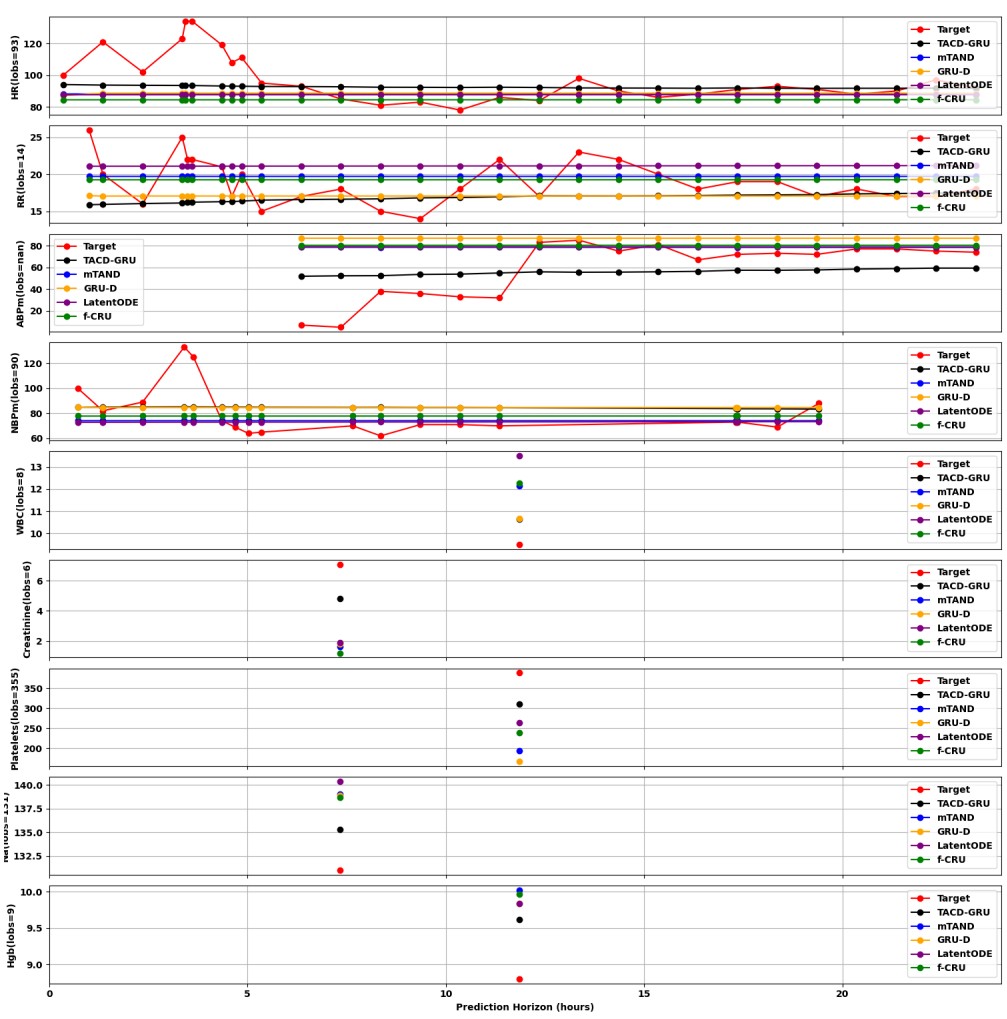

Figure 4: Example 1 for MIMIC-III multi-step prediction task.

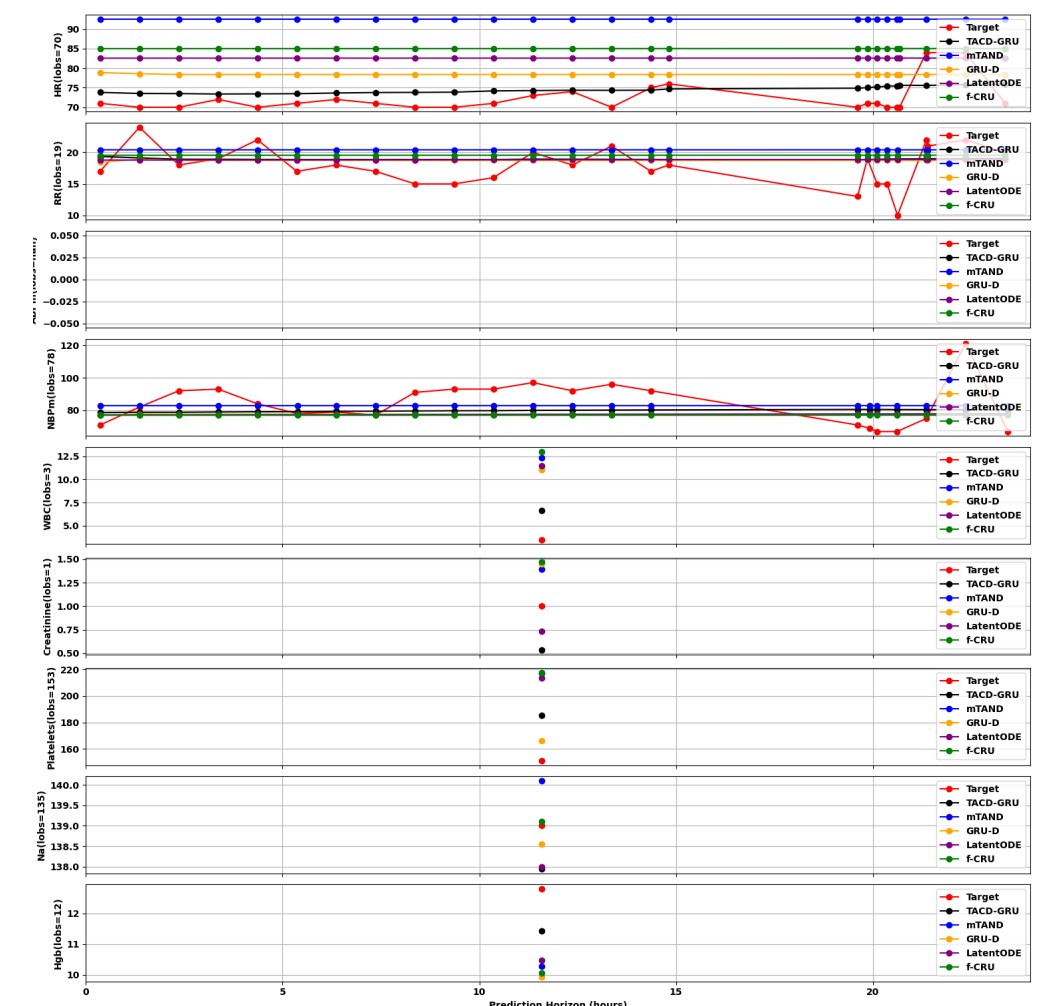

Figure 5: Example 2 on MIMIC-III

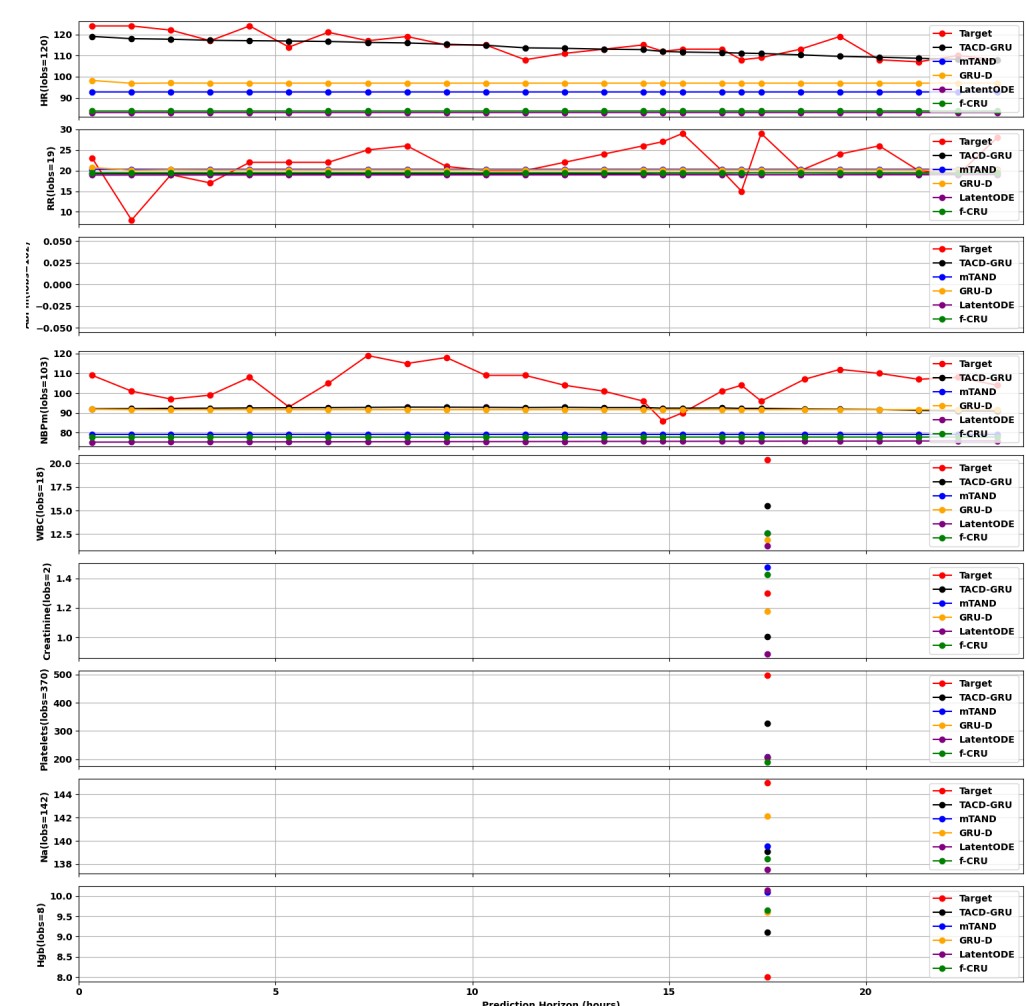

Figure 6: Example 3 on MIMIC-III

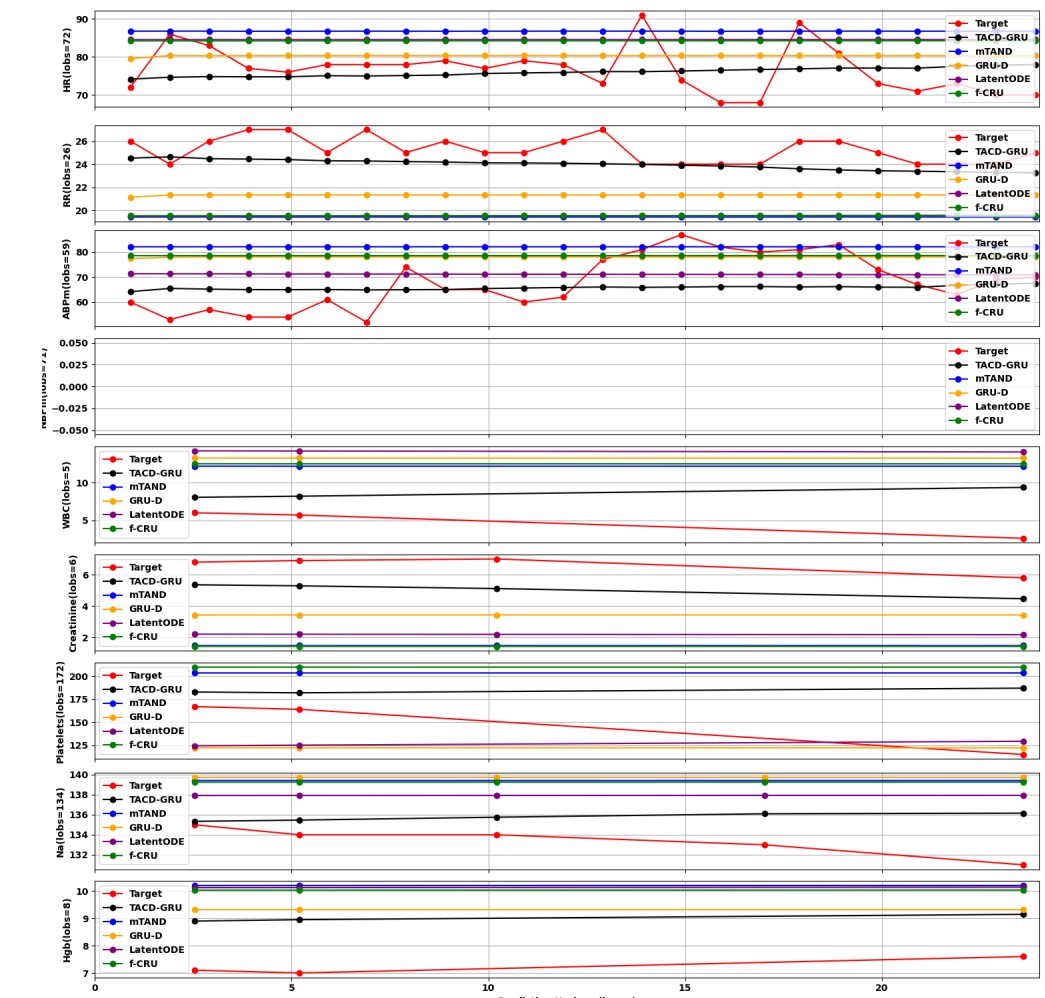

Figure 7: Example 4 on MIMIC-III

## J  TACD-GRU INTERPRETING MIMIC-III EMBEDDINGS

Having trained TACD-GRU on MIMIC-III multi-step prediction, we inspect which variables the attention based model attend to for specific target variable prediction. Since the time-elapsed embedding are concatenated with the time series embedding (as in Equation (12)), we set a fixed univariate target time series and set its time-elapsed to 2 hours. We vary all but the target time series in the range of time elapses 15 minutes to 4 hours in increments of 30 minutes. The choices of these time related parameters are fairly arbitrary, and the main objective is to study the attention of a target variable over other time series at different time-elapses.

We analyze the attention weights for ABPm target over all medication administration time series in Figure 8. Interestingly, there is no effect of medications observed after the last observed target variable indicated by zero-weights until 2 hours of time-elapses (as we fix target's last observation time to 2 hours). The attention based estimator learns the behavior of attending to itself and ignoring other time series if they occur between between the prediction time and its last observed value. It puts non-zero weights to the time series that occur before target's last observed time.

We find that for target ABPm, top-5 medications according to attention weights are "Lansoprazole", "Albumin 25%", "PO Intake" , "OR Colloid Intake" and "Vasopressin". Albumin, PO Intake, and OR Colloid Intake are used for volume (fluid) therapy or cover fluid intake in general and hence, can effect blood pressure. Vasopressin is a medication to treat hypotension (abnormally low-blood pressure condition). It is somewhat unexpected to find Lansoprazole in the top-5 medications. It

is a medication for treating acid reflux, and it is commonly prescribed for ICU patients. However, literature suggests it may have blood pressure reduction effects for hypertensive patient.

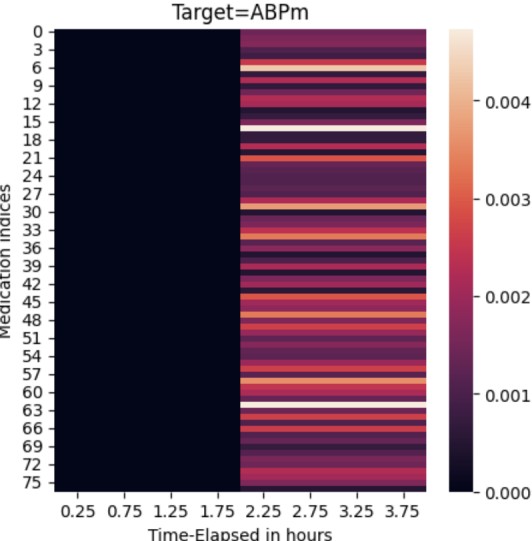

Figure 8: ABPm's attention over medication time series at different time-elapses.

## K ROBUSTNESS OF THE META-DECISION MODEL

Figure 9: Plots for weight assigned to TACD-GRU-ATTENTION at different perturbation noise levels for Physionet (**left**) and USHCN (**right**) datasets. Plots indicate that as TACD-GRU-CONTEXT become more noisier (by increasing the noise level), the meta-decision model assigns more weight to TACD-GRU-ATTENTION. Noise level of 0.0 is no perturbation; 0.25 is 0.25 random noise and 0.75 TACD-GRU-CONTEXT.

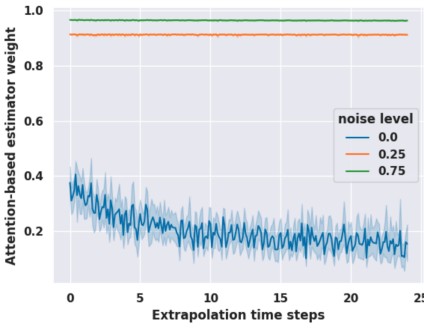 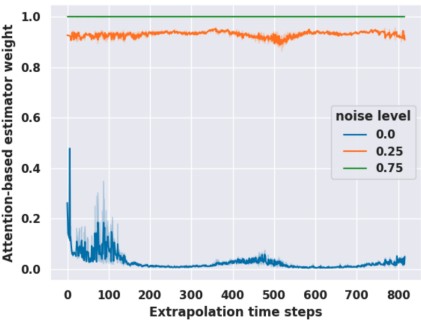

To evaluate the robustness of the meta-model that combines the two estimators, we perturb one of the estimator's predictions by introducing noise. Specifically, we add noise uniformly sampled from the range $[0, 1]$ to the TACD-GRU-CONTEXT predictions. The choice of the range $[0, 1]$ is deliberate because by design, the targets are bounded in the range $[0, 1]$ (due to min-max re-scaling), and consequently, the prediction model should also calibrate to that range. Additionally, to assess the sensitivity of the meta-model at different noise levels, we blend the actual estimates with the noise using a convex combination. The weighting of this combination determines the noise contribution to the final TACD-GRU-CONTEXT's predictions $\widehat{\mathbf{x}}_{t,\Delta T}^{c,n}$ .

$$\widehat{\mathbf{x}}_{t,\Delta T}^{c,n} = c_n \mathbf{n} + (1 - c_n)\widehat{\mathbf{x}}_{t,\Delta T}^c. \tag{16}$$

where each element in the noise vector: $\mathbf{n}_i \sim \mathcal{U}(0, 1)$, and noise coefficient $c_n \in [0, 1]$ is fixed for the entire experiment. For the reported results on robustness, meta model uses $\widehat{\mathbf{x}}^{c,n}_{t,\Delta T}$ instead of $\widehat{\mathbf{x}}^{c}_{t,\Delta T}$.

The expectation is if one of the estimators is susceptible to errors, then the meta-model switches to the other estimator by assigning more weight to it. In our ablation, the TACD-GRU-CONTEXT's prediction is perturbed at different noise levels, and so we expect a robust meta-model to switch to the TACD-GRU-ATTENTION model as noise level (controlled by $c_n$) increase. Figure 9 shows that as we increase the noise, the weight assigned to the TACD-GRU-ATTENTION model progressively increases for both USHCN and Physionet datasets. This significant shift from low to high attention estimator weighting demonstrates the proposed meta-model's ability to dynamically calibrate the model weighting in order to reduce the noise in the final output.

## L  COMPUTATIONAL COST ANALYSIS

We provide a comprehensive comparison of the computational complexity across all models examined in our study in the Figure 10. We evaluate and contrast the training times, inference times, and peak GPU memory usage for each model when performing single-step prediction on the Physionet dataset. All reported times are in seconds and represent averages over all training epochs. Our analysis reveals a clear trend in terms of the train time. We note that methods (ContiFormer, ODE-RNN, Latent ODE) that rely on numerical solvers exhibit the longest training times, followed by models (CRU, f-CRU, RKN-$\Delta_t$) that assume linear dynamics. Recurrent models (TACD-GRU, GRU-D, GRU-$\Delta_t$) demonstrate moderate training times. Lastly, models that can be parallelized in the time dimension (mTAND and T-PatchGNN) are the fastest to train. However, it is crucial to note that the models with the fastest training times (mTAND and T-PatchGNN) come with a significantly higher memory requirement as illustrated in Figure 10b.

Having trained the time series model on retrospective EHR data, our primary goal is to deploy it as an early warning system to prevent adverse patient conditions in the ICU. This model operates in an inherently online context, characterized by a continuous stream of (near) real-time data. This data stream includes critical patient information such as vital signs and medication administration records, which the deployed model processes continuously. To simulate this online environment, we use the Physionet dataset, streaming observations from 100 randomly selected patients in an online manner. The models are evaluated based on total inference time and peak GPU memory usage during inference. In our analysis, we compare TACD-GRU with models that offer the best train times: mTAND and T-PatchGNN. We observe that since mTAND and T-PatchGNN do not have a Markovian latent state representation, it needs to buffer the past observations to make the inference. This results in higher inference cost both in terms of time and memory. Moreover, these costs grows over time. In contrast, models that consists of Markov state representation such as TACD-GRU are independent of the past observations thus, maintaining a constant, low time and memory requirement.

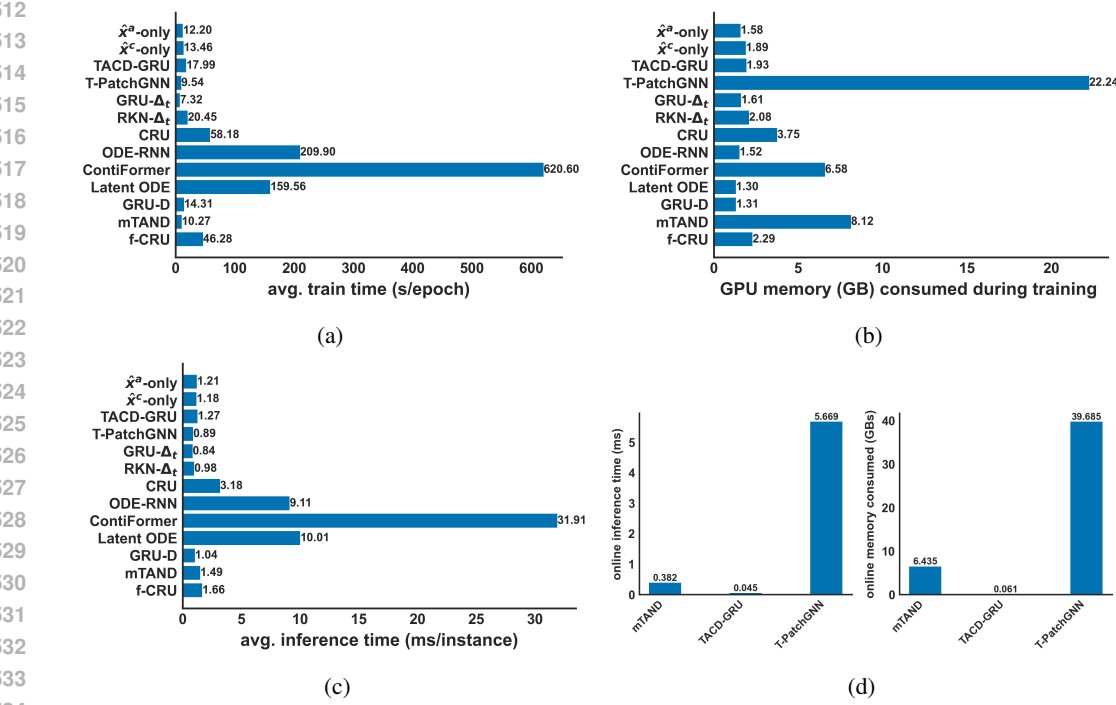

Figure 10: Computational cost analysis on the Physionet dataset. (**a**) Compares the average train time per epoch in seconds. (**b**) Peak GPU memory usage in GBs during training. (**c**) Average inference time in milliseconds per instance. (**d**) Comparison of the inference time (**left**) and memory consumption (**right**) in an online deployment.

## M ADDITIONAL EXPERIMENTAL RESULTS

This section includes additional supporting experimental results for our paper.

Table 5: Additional statistics for MIMIC-III multi-step prediction.

(a) MIMIC-III multi-step prediction TACD-GRU's Win Rate % under MSE and MAE metrics across baseline models.

| Models | Win Rate % | |
|---|---|---|
| | MSE | MAE |
| mTAND | 59.28 | 59.28 |
| GRU-D | 56.70 | 54.90 |
| Latent ODE | 65.21 | 65.72 |
| f-CRU | 64.18 | 65.21 |
| CRU | 63.07 | 61.99 |
| ContiFormer | 67.78 | 69.07 |
| ODE-RNN | 61.46 | 62.80 |
| RKN-$\Delta_t$ | 67.39 | 66.85 |
| GRU-$\Delta_t$ | 64.15 | 63.61 |

(b) MIMIC-III multi-step prediction MSE ($\times 10^{-2}$) analysis on disjoint time intervals in 24-hour prediction window. For instance, $1 - 8$ considers all the targets in prediction horizon range 1 to 8 hours.

| Model | Multi-step pred. ranges (in hours) | | |
|---|---|---|---|
| | $1 - 8$ | $8 - 16$ | $16 - 24$ |
| mTAND | 1.74 | 1.88 | 1.77 |
| GRU-D | 1.35 | 1.54 | 1.50 |
| Latent ODE | 1.69 | 1.88 | 1.78 |
| f-CRU | 1.73 | 1.88 | 1.75 |
| CRU | 1.70 | 1.86 | 1.76 |
| ContiFormer | 1.34 | 1.54 | 1.57 |
| ODE-RNN | 1.55 | 1.73 | 1.63 |
| RKN-$\Delta_t$ | 1.64 | 1.80 | 1.70 |
| GRU-$\Delta_t$ | 1.78 | 1.90 | 1.79 |
| TACD-GRU | 1.13 | 1.41 | 1.38 |
| $\Delta$ over best | (**+0.21**) | (**+0.13**) | (**+0.12**) |

## N COMPUTING INFRASTRUCTURE

We used one server machine to deploy the experiments reported in the paper. This machine is equipped with 100GB memory, one NVIDIA L40S GPU, with Intel Xeon Platinum 8462Y+ @ 2.80 GHz processor and 16 CPU cores.

