# OpenReview forum: "TACD-GRU: Time-Aware Context-Dependent Autoregressive Model for Irregularly Sampled Time Series"
_ICLR.cc/2025/Conference — Submitted to ICLR 2025_

### Official Review · Reviewer_q6Rp · 2024-10-28

**Soundness:** 2
**Presentation:** 3
**Contribution:** 2
**Rating:** 5
**Confidence:** 4

**Summary:**

Forecasting irregularly sampled time series with missing values is a critical yet under-researched area due to the inherent complexities of dealing with both irregular sampling and missing data. This paper introduces a novel model, TACD-GRU, an RNN-based approach designed to forecast irregularly sampled time series with missing values. TACD-GRU incorporates two key mechanisms: (1) a context-aware component that learns from historical observations across the entire timeline, and (2) a time-aware component that focuses on the most recent observations (the last observation). The model was evaluated against a wide range of baselines across three datasets, demonstrating promising results.

**Strengths:**

**S1.** The paper addresses the significant yet under-explored topic of forecasting irregularly sampled time series with missing values. The paper is well-written and easy to follow

**S2.** The approach of forecasting using two complementary modules—context-based and last-observation-based—is interesting

**S3.** The experiments adhered to the existing protocol, and the results demonstrate that the proposed model is promising. In the appendix and code, hyperparameters used for all the competing models are provided

**S4.** A new irregularly sampled time series forecasting dataset from MIMIC-III is useful for the community to further research in this field

**Weaknesses:**

**W1.** Some important literature is missing ([1], [2]). These are graph (attention)-based models designed for forecasting irregularly sampled time series with missing values

**W2.** In terms of modeling, using a two-component model is not new. I see some similarities between SIMTSC [3] and TACD-GRN. SIMTSC also has two components: 1) learning from the context using an RNN, and 2) learning from recent observations (using a GNN). I agree that the specific implementation of each component is different in both models and they are applied in different contexts, but, in my humble opinion, it is necessary to highlight the similarities and distinguish the differences.

**W3.** There are existing datasets from MIMIC-III and MIMIC-IV used in GraFITi [1], GRU-ODE-Bayes and Neural Flows [3]. Instead of using these, why did the authors create a new dataset based on MIMIC-III?

**W4.** Considering the recurrent nature of the model, how efficient is the training process? Could authors provide the runtime and/or evaluation time comparison?



**References:**
1. Yalavarthi, Vijaya Krishna, et al. "GraFITi: Graphs for Forecasting Irregularly Sampled Time Series." AAAI 2024
2. Zhang, Weijia, et al. "Irregular Multivariate Time Series Forecasting: A Transformable Patching Graph Neural Networks Approach." ICML 2024
3. Biloš, Marin, et al. "Neural flows: Efficient alternative to neural ODEs." NeurIPS 2021
4. Zha, Daochen, et al. "Towards similarity-aware time-series classification." SIAM SDM 2022

**Minor:**
- **MW1.** For better clarity, I suggest using $x_{t, \Delta t}$ to represent $x_{t + \Delta t}$ (line 50), and similarly $g_{t, \Delta t}$ for $g_{t + \Delta t}$ (line 279)
- **MW2.** The legend and labels in Figures 2 and 3 are difficult to read

**Questions:**

In addition to the Weaknesses mentioned above

**Q1.** Often the datasets considered here are medical and climate which can have seasonalities. (How) does TACD-GRU incorporate seasonality in the dataset?

**Q2.** Is ${\Delta \tau} = (\tau_t - \tau_{t-1})\cdot \mathbb{1}$ where $\mathbb{1}$ is a column vector with all 1s (lines: 236)

**Q3.** What will be ${x_t^*}$ for $t=0$ (initial value). Assuming it to be a zero vector, can we differentiate observation with a 0 value from the initial value?

**Q4.** While two evaluation metrics, MSE and MAE, are used, for what metric is the loss optimized?

**Q5.** In 4.2 (Model training) it was mentioned that the entire sequence is reconstructed. Is reconstruction loss an auxiliary loss to the forecasting loss or both losses are treated in the same manner?

**Q6.** In Algorithm 1, what is the difference between $\Delta t$ and $\Delta T$

---

### Official Review · Reviewer_RrJD · 2024-11-03

**Soundness:** 2
**Presentation:** 2
**Contribution:** 2
**Rating:** 5
**Confidence:** 3

**Summary:**

The paper proposed a refined RNN-based model designed to predict irregularly sampled multivariate time series. The proposed architecture combines a context-based to capture the long-term dependency and a last-observation-based model that focuses on short-term temporal patterns.

**Strengths:**

The paper tackles an important problem in time-series analysis, as current methods for time-series analysis are mainly based on the assumption of continuous and regular interval observations.
The paper proposes the design of a combination of modeling both temporal and long-term dependency, which could potentially enhance performance in complex time-series scenarios.
The experiment results look nice and insightful, especially the visualizations.

**Weaknesses:**

1. The paper lacks a well-organized structure does not read like a cohesive written paper and does not highlight their motivation and objectives. For example, the way authors describe their methods in the introduction. Also, the module names in intro, method and results are also not consistent.
2. It reads like the paper is just a combination of two modules that lack novelty.
3. The limitation of current literature in multivariate irregular time series is not clear and seems not relevant to what these authors are trying to solve.
4. Miss the important baseline and discussion of irregular time series forecasting problems[1].


[1]Zhang, W., Yin, C., Liu, H., Zhou, X. &amp; Xiong, H.. (2024). Irregular Multivariate Time Series Forecasting: A Transformable Patching Graph Neural Networks Approach. <i>Proceedings of the 41st International Conference on Machine Learning</i>, in <i>Proceedings of Machine Learning Research</i> 235:60179-60196 Available from https://proceedings.mlr.press/v235/zhang24bw.html.

**Questions:**

1. why use 'prediction' rather than 'forecasting' as the term?

---

### Official Review · Reviewer_YqGM · 2024-11-03

**Soundness:** 3
**Presentation:** 3
**Contribution:** 2
**Rating:** 5
**Confidence:** 4

**Summary:**

The paper proposes a new model for the Irregularly sampled multivariate time series prediction task combining the benefit of RNNs and attention mechanism.  The GRU-based asynchronous RNN unit forms the context based model while attention over the last observations focuses on short term dependencies.

**Strengths:**

Handling sporadically observed time series is an important research topic that helps unlocking the full potential of time series modeling for practical problems. The present method combines attention with RNN based model and time decaying hidden states.  The paper provide detailed ablation studies to show the benefits of these components and their combination.

**Weaknesses:**

Some of the standard deviations are extremely high in the tables, it is clear that executing only 3 repeats is not enough.
In case of shortage of available computational  power, please at least execute  more repeats on cases when there is extreme std (within an order of magnitude to the mean) and/or they are in the TOP of the list.
Just for example: On Table 1 you cannot bold Contiformer in the first column without bolding RKN-Delta_t and mTAND as well as it stands now.

Appendix B mentiones that special effort was made to stay comparable with Schirmes et al.
It is very hard to compare the evaluation of different papers even if it was carried out on the same dataset with same methods. I quickly checked the intersection of methods on the present work, Schirmes et al.  and De Brouwer et al.
8 method was tested in common with Schirmes et al.

| MODEL      | Present paper 1step | Present paper multistep | De Brouwer et. al. | Schirmes et al. Extrapolation |
|------------|---------------------|-------------------------|--------------------|-------------------------------|
| Latent ODE | 0.6 (0.2)           | 152.3 (1.7)             | 0.96 (0.11)        | 203.4 (0.5)                   |
| GRU-D      | 1.5 (0.9)           | 142.8 (19.8)            | 0.53 (0.06)        | 171.8 (1.5)                   |
| GRU-ODE-B  | n.a.                | n.a.                    | 0.43 (0.07)        | 543.7 (102.0)                 |
| GRU        | n.a.                | n.a.                    | 0.75 (0.12)        | 207.1 (1.51)                  |
| ODE-RNN    | 1.9 (1.7)           | 172.4 (1.9)             | n.a.               | 195.5 (46.6)                  |
| CRU        | 3.0 (1.9)           | 136.1 (27.5)            | n.a.               | 127.3 (6.6)                   |
| f-CRU      | 2.0 (0.7)           | 161.1 (5.1)             | n.a.               | 156.9 (32.1)                  |
| mTAND      | 0.9 (0.6)           | 159.3 (1.7)             | n.a.               | 236.0 (3.8)                   |
| RKN-dT     | 0.6 (0.4)           | 146.1 (21.9)            | n.a.               | 149.1 (27.2)                  |
| GRU-dT     | 3.5 (0.1)           | 170.1 (7.0)             | n.a.               | 208.1 (5.4)                   |

(Here i assumed the most likely interpretation of “MAE (×10−2) ”.

Just as a quick check I calculated Spearman rank correlatinon of the 2nd and 4th column, and I got 0.57, with not significant (0.1389) p-value. So without more standardization of the experiments, we cannot even decide the order between methods (on the same dataset!). I have to note here that it is not the job of the present authors to solve this issue of the field, but currently the best I can assume is that the method is in the legue of SOTA.

Relative to Schirmes et al, you do not compare to RKN, GRU and GRU-ODE-Bayes. Why leaving out this three? IN case of GRU and RKN we can argue that they tend to be always worse than the $\Delta t$ versions.

Minor:
Clarify notation, for example in Eq. 9,  the formula for $g_{t, \Delta T}$ contains $\Delta\tau$

“the most set of recently observed values for all variables,” -> set of most recently?

**Questions:**

1) Please execute more repeats when it is necessary! (see above)

2) "One limitation of ODE-based methods is that it reaches to a solution as a function of initial condition. However, the initial condition cannot be adapted to the observed distribution. NeuralControlled Differential Equations Kidger et al. (2020) is proposed to address this limitation. " ← I do not really understand the statement here? In any case the initial condition can be taken as a variable and back propagated to. In case of Neural control DEs as well as in case of GRU-ODE-Bayes jumps are possible so observational information can overwrite the past and make the initial condition irrelevant. What is special about NCDEs? Can youl clarify?

3) How this should be interpreted : xMSE (×10−2) and MAE (×10−2) ?
  /The values in the table are already multiplied by  10-2 so proper value is say 0.020 -> 2.0, or the content of the table have to be multiplied with 10-2 so the proper value is 0.02 -> 0.0002 ?

---

### Official Review · Reviewer_Wr5e · 2024-11-04

**Soundness:** 3
**Presentation:** 3
**Contribution:** 2
**Rating:** 6
**Confidence:** 3

**Summary:**

This work proposes a dual model named TACD-GRU, which combines long-term context and short-term last observations for irregularly sampled multivariate time series prediction. For long-term context, TACD-GRU uses GRU to update the hidden state given an observed value and state decays with time between observations. For short-term observations, an attention mechanism is used to summarize last observations over all variates. These two prediction are combined by the weight output by an MLP. Single- and multi-step experiments on three real-world datasets demonstrate the effectiveness of the proposed model.

**Strengths:**

1. The paper is well-structured and clearly written, making it accessible to readers.

2. The TACD-GRU model presents a straightforward approach by combining context-based and attention-based predictions.

3. The paper includes extensive empirical evaluations across multiple real-world datasets, demonstrating the model's superior performance compared to baselines.

**Weaknesses:**

1. Despite the simplicity, the context-based part is lack of novelty. The use of GRU to model abrupt changes and continuous processes to model slow variations between observations has been extensively studied in previous works[1,2]. Espscially for event sequence works[3,4], which exaclty use GRU and exponential decay.

[1] Neural Jump Stochastic Differential Equations. In NeurIPS-2019.

[2] GRU-ODE-Bayes: Continuous modeling of sporadically-observed time series. In NeurIPS-2019.

[3] The Neural Hawkes Process: A Neurally Self-Modulating Multivariate Point Process. In NeurIPS-2017.

[4] Neural Relation Inference for Multi-dimensional Temporal Point Processes via Message Passing Graph. In IJCAI-2021.

2. This paper lacks an evaluation of efficiency and complexity. Since the proposed method is based on RNNs, it cannot be parallelized, and the attention mechanism grows quadratically with the number of variates. Therefore, such analysis and evaluation are necessary.

**Questions:**

See the #Weakenesses section.

---

### Meta-Review · Area_Chair_kMoh · 2024-12-21

**Metareview:**

This paper  presents a new RNN-based model for Irregularly sampled multivariate time series forecasting that combines a context-based and an attention-based mechanisms. Reviewers appreciated the modeling approach and the empirical evaluation. However there were concerns around the experimental results which weakens the results of the paper - especially in terms of the high standard deviation and important recent missing baselines (Grafiti and T-PatchGNN), and the reasoning behind deriving a different version of the MIMIC datasets. While the authors did add more baselines and experiment seeds during the rebuttal process, the updated results, especially when compared to Grafiti do not showcase a consistent significant improvement over the state-of-the-art. I would urge the authors to conduct a more exhaustive evaluation of their model and resubmit to a future venue.

**Additional Comments On Reviewer Discussion:**

Most reviewers had questions around the model missing two important recent baselines (Grafiti and T-PatchGNN), around the high standard deviations of the reported experimental results, and around the computational costs of the model. While the authors updated their experiments to address the above points, the resulting comparisons against the recent baselines (especially Grafiti) did not demonstrate a consistent significant improvement.

---

### Decision · Program_Chairs · 2025-01-22

Reject